# Perceptions and Attitudes of Generation Z Students towards the Responsible Management of Smart Cities

Sorin-George Toma [1], Cătălin Grădinaru [1], Oana-Simona Hudea [1] and Andra Modreanu [2,*]

1    Faculty of Administration and Business, University of Bucharest, 030018 Bucharest, Romania; sorin.toma@faa.unibuc.ro (S.-G.T.); catalin.gradinaru@faa.unibuc.ro (C.G.); oana.hudea@faa.unibuc.ro (O.-S.H.)

2    Doctoral School of Business Administration, Bucharest University of Economic Studies, 010374 Bucharest, Romania

\*    Correspondence: andra.modreanu@faa.unibuc.ro

**Abstract:** The emergence and development of smart cities represent a significant challenge for the post-modern world. Generation Z members currently entering adult life will play an important role in the implementation of the concept of a smart city. The objective of this study is to identify and analyze Generation Z students' perceptions and attitudes towards the responsible management of smart cities. Following a quantitative approach, the authors designed and applied an online survey in order to reach the purpose of the study. The research sample comprised 468 representatives of Generation Z final-year undergraduate students at a university located in a smart city in the making, Bucharest (Romania). The data were statistically analyzed and interpreted using various tools and methods, such as Cronbach's alpha and correlational analysis. The results show that students are aware of the role played by the city government in ensuring responsible management of the economic, social, and environmental issues of a smart city. Also, they emphasize that most of them are involved in or would like to be involved in different projects specific to smart cities. These results may represent the starting point for understanding Generation Z students' expectations regarding responsible management in the context of smart cities.

**Keywords:** smart city; responsible management; Generation Z; students; perceptions; attitudes

## 1. Introduction

Nowadays, human society is, to a large extent, urbanized [1]. In 2021, around 57% of the world's population lived in urban areas [2]. Since entering the modern century, humanity has faced a plethora of opportunities and threats. These challenges have raised numerous economic, social, and environmental problems for the post-modern world. This is why the emergence and development of smart cities and the implementation of their principles in various domains, such as transportation and construction, have constituted an innovative and intelligent solution for policymakers, city government, and urban management [3].

The basis of smart city expansion is a responsible management approach that ensures their effective functioning in the economic, social, and environmental domains [4]. So far, management decisions in the context of smart cities have been related to the following six pillars: smart economy, smart people, smart environment, smart governance, smart living, and smart mobility [5]. Because local community involvement plays an essential role in sustaining these fundamental areas of activity in smart cities, Generation Z's (Gen Z's) potential contribution to the development and implementation of such initiatives was acknowledged by several authors due to the characteristics of its representatives [6–8].

In this respect, Gen Z's perceptions and attitudes constitute a topic of interest for researchers, especially those of young students. Attitude is defined as a mental state

shaped by experience, which influences one's reaction or response to all objects and circumstances associated with it [9], while perception is described as a process that enables individuals to interpret and, therefore, to make sense of the received information or experienced feelings [10]. In comparison with other generations, Gen Z students were raised in a highly connected and digital environment, and as a consequence of this fact, they are open to embracing rapidly advancing technologies in order to enhance their urban experience [11–14].

Moreover, representatives of this generation that are now entering adult life are perceived as being broadly oriented and adaptable due to the fact that they are interested in various topics related to environmental, social, and economic issues [15]. As a consequence, Gen Z students are considered to be global thinkers and changemakers [16]. They are conscious of current and emerging global challenges and willing to act responsibly to solve economic, environmental, and social issues [15]. In addition, Gen Z students are characterized as being equitable, altruistic, and socially conscious due to their eagerness to contribute to the well-being of society in general [17]. Apart from their passion for Information and Communication Technologies (ICT), they share other common characteristics that are related to the dimensions of responsible management of a smart city as follows [8,18–20]: finding various job opportunities; seeking well-paid jobs; benefiting from good health services; facing less corruption; benefiting from good education; social involvement; environmental consciousness, etc. Additionally, the results of several studies sustain that, in general, Gen Z students have a positive attitude towards the concept of a smart city, perceiving it as an opportunity for sustainable and responsible development of society [21]. However, not all Gen Z members are highly familiar with this concept, as some of them are not acquainted with the concept and its principles. This is why our research was focused on Gen Z students' perceptions and attitudes towards the responsible managerial decisions of smart cities.

Thus, the three research domains outlined before—smart city, responsible management, and Gen Z—constituted the basis of the literature review deployed by the authors. Additionally, starting from the above-mentioned theoretical framework, the authors established the following research questions:

1. How does the social orientation of a smart city influence Gen Z students' perceptions and attitudes towards its responsible management?
2. How does the environmental orientation of a smart city influence Gen Z students' perceptions and attitudes towards its responsible management?
3. How does the economic orientation of a smart city influence Gen Z students' perceptions and attitudes towards its responsible management?
4. How do the managerial decisions of a smart city determine whether students are involved in social, environmental, and economic projects initiated by the authorities of the city and NGOs?

The research presented in this article seeks to fill the highlighted research gap and provides new possible directions for other researchers. The objective of this study is to identify and analyze Gen Z students' perceptions and attitudes towards the responsible management of smart cities. The results of this study show that final-year undergraduates of business and administration studies in the 21–23 age group, representatives of Gen Z, are aware of the importance of responsible management in the context of smart cities. Also, it emphasizes their desire to be involved in different projects, such as social, economic, and environmental projects, designed to expand the use of the principles of smart cities. The paper is structured as follows: The next section presents the literature review. Materials and Methods are displayed in Section 3. Sections 4 and 5 present the results of the research and discussion. The last section illustrates the conclusions of the paper.

## 2. Literature Review

### 2.1. Smart City: Definition and Dimensions

After its first use in the 1990s [22], the smart city concept was often presented as having a sustainability-oriented approach ([23], p. 5). The sustainability concept often depicts aspects such as ensuring social equity, the conservation of the natural environment and bio-diversity [24], maintaining social-economic vitality [25], and the quality of life in the urban environment, factors that are also considered goals for a smart city. A smart city should be designed to be very responsive, having the capacity to rapidly adapt to various changes, such as technological development, which is often disruptive. As such, there are smart city definitions that focus on the ICT aspects, highlighting that mobile devices, edge-cloud computing [26], and the Internet of Things [27] are important tools for the managers of smart cities [14] and for smart city information processing [28].

The focus of a smart city is placed on the vital role played by ICT infrastructure [29], using a mixture of state-of-the-art technologies [30], information [31], and human, social, and relational capital alongside environmental interest to pursue citizen benefits such as reducing costs, optimizing resource consumption, improving interactivity, and enhancing the quality of life and overall people's welfare [32–35]. These expected results address issues concerning economic, social, and environmental pillars and require combined socio-technical efforts [36] and a rational management of natural resources [37] to ensure an interconnected, secure, attractive, and sustainable community [38]. Thus, a smart city is connected to three main pillars of sustainability: environmental, economic, and societal.

Linked to the smart city concept, there are several initiatives that fall under six dimensions: economy, governance, environment, mobility, living, and people [29,39].

Smart Economy, a concept also linked to policy [40] means pursuing opportunities using ICT, implying e-business and e-commerce, smart business processes, smart technology services, and even new smart business models.

Smart Environment entails the implementation of smart resource management to create a space for ecology and biodiversity to coexist within the urban environment [41]. It is comprised of the smart energy management of [42] energy grids or renewable energy sources, smart energy data analytics, energy security, or energy prosumers and consumers [43] that should lead to the development of a sustainable smart energy city [44]. Green urban planning and design have led to the development of green buildings that play an important role in managing energy consumption and reducing $CO_2$ emissions [45]. A smart city, through its green infrastructure, shapes a public open space landscape that generates cognitive and restorative benefits for its residents [46].

Smart Living revolves around the integration of advanced technologies in cities and homes [47] and refers to initiatives that use smart technology under sustainable conditions to facilitate intelligent living, enabling improved or new lifestyles that enhance the quality of living and provide a safe and healthy city for its inhabitants [48]. The scope is to enhance the living experience of a city's inhabitants regardless of age or other demographics as it proposes innovative solutions that make life sustainable, efficient, integrated, controllable, productive, and economical [49].

Smart Mobility or movement is the concept that shapes the transportation of people and goods and the planning of cities using ICT [50]. Thus, it describes a multitude of mobility options and services that are provided by the internet, telecommunications, and technology [51], managing to solve navigation problems such as congestion or pollution [52]. The specific initiatives of smart mobility are aimed at reducing pollution, traffic congestion, people's safety, noise pollution, transfer costs, increasing people's safety, and improving transfer speed [53].

Smart People/society is highly relevant in a knowledge-based economy, as this dimension refers to those who master skills related to information and economy [54]. This dimension is linked to improving creativity and fostering innovation through ICT and requires creative, adaptable, responsible, and productive citizens [55]. The use of ICT enables work-from-home, capacity management, or access to training and education. Within a

smart city, superior social interactions and relationships are formed to connect residents amongst themselves and to the outside world [56].

Smart Governance implies the use of ICT to provide promising solutions using innovative services to the inhabitants of a smart city [57]. It refers to the technology-enabled collaborations involving local governments and citizens to promote advancements in the field of sustainable development [58]. There are three main domains in which the concept operates, namely public services, bureaucracy, and public policy [59].

Each dimension may ensure competitiveness (smart economy), the development of social and human capital (smart people), a high level of participation (smart governance), better transport and ICT (smart mobility), efficient use of natural resources (smart environment), and a high quality of life (smart living) [60].

Starting from its dimensions, a smart city needs to identify and implement a type of management that ensures, among other things, its sustainability, efficiency, and effectiveness. One of them is responsible management.

### 2.2. Responsible Management in the Context of Smart Cities

Since the end of the Second World War, responsibility has become a topic of interest for both theoreticians and practitioners all over the world. Centered mostly on the concept of stakeholders, the issue of responsibility has been primarily addressed in the business domain.

In the 1950s, organizational responsibility was considered "an obligation to pursue those policies, to make those decisions, or to follow those lines of action that are desirable in terms of the objectives and values of our society" ([61], p. 6). Later, it was defined as "context-specific organizational actions and policies that take into account stakeholders' expectations and the triple bottom line of economic, social, and environmental performance" ([62], p. 858).

Important for any type of organization—either government, public institution, or corporation [63]—responsible management may be defined as the "process involving tools for managing social, environmental, and economic capital and impact throughout activities and functions" ([64], p. 75). It embeds organizational responsibility, environmental sustainability, and ethics into organizational practices [65] and aims to optimize overall stakeholder value. Thus, from a managerial perspective, responsible management refers to assumed responsibilities towards stakeholders. Its transdisciplinarity derives from the integration of its three dimensions: sustainability, ethics, and responsibility. Deeply related to social, economic, and environmental issues, sustainability is basically based on the concept of the triple bottom line, while ethics is connected with morality and addresses the issues of right and wrong.

With an intense individual-level accent, responsible management puts the manager at its center. This focuses on the way a manager takes into account not only his experience, abilities, and competencies but also his behavior and role. In other words, responsible management involves praxis, or what a manager does in his/her everyday activities [66]. In the case of a smart city, he/she should identify the various stakeholders (e.g., institutions, companies, citizens, and non-governmental organizations) and meet their needs and expectations [67].

A smart city is not limited to successfully implementing various technologies but also represents an approach that identifies and uses the most appropriate model for its management. As a broad strategic concept, it embraces technology and innovation on a large scale and continuously improves the lives of its citizens by addressing socio-economic and environmental issues. Among other things, a smart city should be livable, digital, safe, sustainable, and efficient. This is why responsible management has proven to be one of the best ways to govern a smart city.

By intensely using technology in all their activities and processes, smart cities impose on their citizens the obligation to be not only creative and democratic participants but also frequent ICT users. In this respect, young people composing Gen Z are highly familiar

with ever-changing digital technologies and the Internet. Being part of the most connected and technology-based generation, their perceptions and attitudes toward the responsible management of a smart city are worthwhile.

### 2.3. Generation Z: Definition and Characteristics

Social, environmental, economic, and technological changes that occur in a given period of time or location can create distinct experiences for different groups of people, and at the same time, these changes can produce similar experiences for those within a given group [6]. Consequently, a generation is defined as "a distinguishable category of people that to some extent have in common aspects such as year of birth, age, physical space, and major life events that occurred during significant developmental phases" ([68], p. 9).

As the first generation born in the world of "Internet-connected technology", technological innovation has played a defining role in the life of Gen Z ([14], p. 3). Nevertheless, the authors did not pinpoint a certain period of Gen Z representatives' birth [69]. Some suggest that post-millennials were born after 1995, while others state that Gen Z members can be considered those who were born after 1997. For instance, according to the Pew Research Center [70], individuals so far can be grouped into five generations, namely: the Silent (consisting of those born between 1928 and 1945, therefore, ages 78 to 95 in 2023); the Baby Boomers (consisting of those born between 1946 and 1964, therefore, ages 59 to 77 in 2023); the Generation X (consisting of those born between 1965 and 1980, therefore, ages 43 to 58 in 2023); the Generation Y likewise widely referred to as Millennials (consisting of those born between 1981 and 1996, therefore, ages 27 to 42 in 2023); the Generation Z generally known as Post-Millennials (consisting of those born between 1997 and 2012, therefore, ages 11 to 26 in 2023) and the Alpha (consisting of those born from 2013 onward, therefore, ages 10 and below). The six generations defined by the Pew Research Center are shown in Figure 1. The present paper focuses on identifying the particularities of Generation Z members.

Regardless of one's viewpoint on the term "post-millennials", it is certain that Gen Z representatives are currently young individuals, namely children of primary schools, teenagers of high schools, and young adults of undergraduate or postgraduate universities [71]. Several authors stated that what characterizes post-millennials is that they are open and eager to use current and new technologies [72,73]. Moreover, in the literature, Gen Z's interests are described as being various. For instance, Gen Z members often participate in social activities related to environmental protection, renewable energy, and sustainable development due to their interest in climate change and overexploitation of natural resources [74].

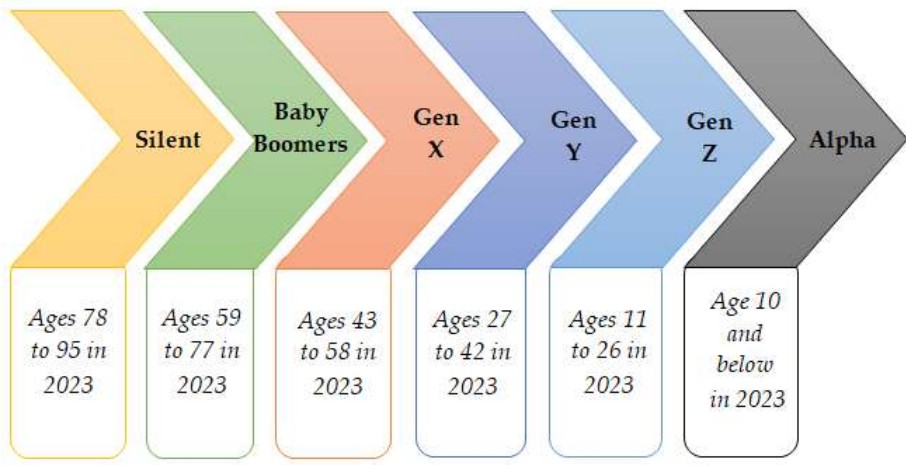

**Figure 1.** The generations defined. (**Source**: Authors' representation).

Gen Z representatives understand that sustainable development requires a group effort, and therefore, they have a positive attitude towards volunteerism and community service [75]. Consequently, they are open to working with businesses, governments, and NGOs to find new and innovative solutions to address environmental and social concerns [76]. Compared to other generations, Gen Z is more inclined to adopt sustainable practices, such as using eco-certified products, using alternative transportation like bicycle sharing, and reducing waste through recycling [77]. As a consequence, its members hold a favorable view of businesses that prioritize ethical and sustainable practices for economic development [78].

Although Gen Z representatives have a global perspective, they are also interested in participating in community development projects [79]. This is because they see social programs aimed at improving the well-being of communities as meaningful and purposeful, with the ability to make a positive impact on quality of life [80]. Moreover, post-millennials frequently participate in social projects focused on social justice and inclusion [81].

What additionally characterizes Gen Z is that they have an entrepreneurial mindset and an innovative approach to change [82]. Due to the fact that they have an innate comprehension of ICT, Gen Z representatives have the ability to swiftly integrate new technologies into their business practices [83]. Consequently, this may lead to business growth opportunities for these young entrepreneurs as they are able to reach a broader audience for their businesses through the use of digital tools and platforms [84]. Nevertheless, post-millennials are open to receiving guidance and support in terms of business management from others, and therefore, public affairs networking and organizations such as business accelerators or incubators are seen as opportunities to develop business skills [85].

## 3. Materials and Methods

To achieve the research objective, namely to identify and analyze Gen Z students' perceptions and attitudes towards the responsible management of a smart city, the authors set up the next research hypotheses:

**Hypothesis 1 (H1).** *There is a significant positive correlation between the social and strategic orientation of the responsible management of a smart city and the decisions regarding social services and housing.*

**Hypothesis 2 (H2).** *There is a significant positive correlation between the social and strategic orientation of the responsible management of a smart city and the decisions regarding environmental management.*

**Hypothesis 3 (H3).** *There is a significant positive correlation between the social and strategic orientation of the responsible management of a smart city and the decisions regarding entrepreneurship.*

**Hypothesis 4 (H4).** *There is a significant positive correlation between the environmental and economic orientation of the responsible management of a smart city and the decisions regarding anti-pollution actions and environmental management.*

**Hypothesis 5 (H5).** *There is a significant positive correlation between the environmental and economic orientation of the responsible management of a smart city and the decisions regarding digitalization.*

**Hypothesis 6 (H6).** *There is a significant positive correlation between the environmental and economic orientation of responsible management of a smart city and decisions regarding entrepreneurship.*

**Hypothesis 7 (H7).** *There is a significant positive correlation between the innovativeness orientation of the responsible management of a smart city and the decisions regarding health and education.*

**Hypothesis 8 (H8).** *There is a significant positive correlation between the innovativeness orientation of the responsible management of a smart city and the decisions regarding anti-pollution actions and environmental management.*

**Hypothesis 9 (H9).** *There is a significant positive correlation between the innovativeness orientation of the responsible management of a smart city and the decisions regarding digitalization.*

**Hypothesis 10 (H10).** *The characteristics of the smart city and the specific social management decisions positively influence the students' desire to get involved in social ICT-based projects initiated by the authorities of the city and NGOs.*

**Hypothesis 11 (H11).** *The characteristics of the smart city and the specific environmental management decisions positively influence the students' desire to get involved in environmental ICT-based projects initiated by the authorities of the city and NGOs.*

**Hypothesis 12 (H12).** *The characteristics of the smart city and the specific economic management decisions positively influence the students' desire to get involved in economic ICT-based projects initiated by the authorities of the city and NGOs.*

Furthermore, to attain the goals of the paper, the authors pursued a specific scientific research methodology. In the beginning, they designed the methodological process. Then, the authors carried out a comprehensive literature review. They identified the sources of secondary data (e.g., articles, books) from several domains (e.g., management, social responsibility) through desk research. The information was collected mainly from electronic databases (e.g., SAGE, Springer) and libraries (e.g., the Central University Library Carol I of Bucharest, the Romanian National Library). After that, the authors categorized, analyzed, and synthesized the data. Then, they formulated the questionnaire (Appendix A) based on the literature review. It was a structured questionnaire designed for quantitative research that consisted of closed questions with single and multiple-response questions. Questions were both behavioral and attitudinal in order to understand what final-year undergraduate students do and think in relation to the responsible management of a smart city. The questionnaire comprised 16 questions, grouped as follows:

- One filter question regarding the year of birth of the respondents (to identify the specific respondents, namely Gen Z students);
- Seven issues regarding Gen Z students' perceptions and attitudes toward the responsible management of a smart city that encompassed its features and the three main managerial orientations (e.g., social, environmental, and economic);
- Eight questions related to socio-demographic data (e.g., gender, professional status, field of activity, field of study (high school), specialization, place of birth, place of residence, net income).

In order to measure the multi-item concepts, a five-point Likert scale (1 = strongly disagree and 5 = strongly agree) was used. Moreover, the authors selected the target population from the field of higher education as final-year students belonging to Gen Z. The authors chose all five undergraduate programs within a public higher education institution located in the capital of Romania, entitled the Faculty of Business and Administration, University of Bucharest, due to the following main reasons:

- The target population (Table 1) comprised only final-year undergraduate day-course students aged 21–23 years, ensuring, therefore, its representativeness. Respondents were Gen Z members.
- There are only two public higher education institutions in Bucharest that provide both Business and Administration specializations (the University of Bucharest and the Bucharest University of Economic Studies). The total number of final-year undergraduate day-course students is around 2200.

- The final-year undergraduate students represent important stakeholders in smart cities as they are tomorrow's businessmen, entrepreneurs, and public servants. Some of them were already employed.
- All undergraduate students were familiar with the key concepts of the paper (smart city, responsible management) as they studied them in numerous topics, such as Management, Public Management, Public Sector Economics, Informatics in Public Administration, Public Policy, Social Policy, or Socio-Economic Phenomena Modeling, within their specializations. First- and second-year undergraduate students did not fully acquire the basic knowledge of the concepts of smart cities and responsible management.
- The size of the target population made possible the use of a mix of research methods, specifically exploratory and descriptive.
- The sampling was made only of students who attended final-year undergraduate programs.
- As its size was relatively small, the sample comprised the whole population of final-year undergraduate students. The respondents were males and females, as no one declared being non-binary (Table 2).
- The authors are teaching different disciplines to students from all five specializations.

**Table 1.** The total number of final-year students and their gender within the undergraduate programs.

| Specialization | Number of Final-Year Students | Gender | |
|---|---|---|---|
| | | Male | Female |
| Business Administration (in Romanian) | 172 | 83 | 89 |
| Business Administration (in English) | 54 | 24 | 30 |
| Marketing | 106 | 41 | 65 |
| Economic Cybernetics | 75 | 49 | 26 |
| Public Administration | 211 | 72 | 139 |
| Total | 618 (100%) | 269 (43.53%) | 349 (56.47%) |

**Table 2.** The total number of respondents and their gender within the undergraduate programs.

| Specialization | Number of Respondents | Gender | |
|---|---|---|---|
| | | Male | Female |
| Business Administration (in Romanian) | 130 | 63 | 67 |
| Business Administration (in English) | 42 | 18 | 24 |
| Marketing | 80 | 31 | 49 |
| Economic Cybernetics | 57 | 37 | 20 |
| Public Administration | 159 | 55 | 104 |
| Total | 468 (100%) | 204 (43.59%) | 264 (56.41%) |

In order to validate the hypotheses of the study, the authors used a quantitative research method, namely an Internet-based survey. The fieldwork research was organized in the period 1–27 April 2023. All the research hypotheses were tested through the application of an online questionnaire to the whole population of final-year students within the undergraduate programs, and their participation was voluntary. A total of 468 questionnaires were validated from final-year (third-year) students (150 out of 618 students were not Gen Z members, sent incomplete responses, or did not answer), with a response rate of 75.72%, proving the quality of the survey [86]. Most of the respondents were female (56.41%), in accordance with the gender structure of the total number of final-year students (56.47%). A smaller sample size, as in this research, led to a higher response rate and reliability of the data [87]. The data obtained online were centralized and systematized in a database and further processed with SPSS statistical software (Version 23, IBM, New York, NY, USA).

They were preceded by the delimitation of the related methods, keeping in mind the hypotheses to be tested.

The first step, given the quite large number of items grouped around four variables, namely: (1) potentially necessary characteristics of smart city-centered responsible management (16 items); (2) specific social issue-related management decisions potentially beneficial for a smart city (13 items); (3) environmental management decisions potentially beneficial for a smart city (15 items); and (4) economic aspect-related management decisions potentially beneficial for a smart city (11 items), was to use the Cronbach's Alpha reliability test so as to check the consistency of the items in relation to the said scale.

Concretely, the Cronbach's Alpha coefficient takes over the score of each item, correlating the same with the total score of each observation, and thereafter the arising correlations are compared as against the variance of all individually computed item scores.

Provided that this test is validated, the condensation of such items into specific constructs is performed by resorting to Principal Components Analysis (PCA), a process that decreases the dimensionality of a dataset by converting some correlated items via orthogonal transformation into a lower series of linearly uncorrelated ones, usually with eigenvalues exceeding 1. PCA consists of several steps, comprising the standardization of items, the computation of the covariance matrix, the determination of the eigenvalues and of the explained variance, and, finally, the identification of the principal components to be considered out of the total range.

Once the principal components are delimited, correlational and regressive analyses are facilitated.

Given that the authors deal with ordinal data, the well-known Pearson coefficient should be avoided, and in exchange, they should resort to other specific correlational techniques, such as the directional Spearman coefficient and Kendall coefficient, both of which deal with ranks and provide information relating to the bivariate correlation between variables.

As regards the regressive analysis, multiple linear regressions, considering the effect of several explanatory variables on the dependent one, are tested herein.

$$y_i = \beta_0 + \beta_1 x_{i1} + \beta_2 x_{i2} + \ldots + + \beta_p x_{ip} + \varepsilon$$

with $y_i$ representing the dependent variable, $x_i$, the independent ones, $\beta_0/\beta_p$, the constant/slope coefficients for the independent variables, and $\varepsilon$, the error term.

All along the quantitative approach, the authors resorted to SPSS, the parametric and non-parametric analysis software usually used for carrying out this particular type of research.

## 4. Results

After having identified the variables encompassing the items of interest, followed by an adequate collection of related data and the identification of the proper methods to be called in the related context, the authors implemented, one at a time, the above-mentioned steps.

The results for the first group of items (1), validated for all 468 observations considered, are presented in Table 3, rendering the reliability statistics with a Cronbach's Alpha coefficient of 0.888, indicating, by virtue of a value superior to 0.7, a high internal consistency of the analyzed items in the related scale; the item total statistics display the scale mean, variance, and Cronbach's Alpha if the related item is deleted, certifying the serious contribution of each item belonging to group (1) to the overall consistency, no item being removable from the list.

**Table 3.** Reliability and Item-Total Statistics.

| Cronbach's Alpha | Cronbach's Alpha Based on Standardized Items | | | N of Items |
|---|---|---|---|---|
| 0.888 | 0.886 | | | 16 |
| **Item** | **Scale Mean if Item Deleted** | **Scale Variance if Item Deleted** | **Corrected Item-Total Correlation** | **Cronbach's Alpha if Item Deleted** |
| Transparent | 61.12 | 65.914 | 0.639 | 0.878 |
| Participative | 61.12 | 66.835 | 0.567 | 0.880 |
| Proactive | 61.53 | 63.770 | 0.743 | 0.873 |
| Socially responsible | 61.20 | 65.790 | 0.579 | 0.880 |
| Open to new | 61.15 | 67.449 | 0.512 | 0.882 |
| Innovative | 61.31 | 69.457 | 0.358 | 0.889 |
| Flexible | 61.32 | 66.535 | 0.559 | 0.881 |
| Equitable | 61.71 | 64.236 | 0.699 | 0.875 |
| Ethic | 61.43 | 64.044 | 0.729 | 0.873 |
| Visionary | 61.22 | 69.908 | 0.345 | 0.889 |
| Decision-oriented | 61.17 | 69.875 | 0.343 | 0.889 |
| Economically efficient | 61.14 | 70.110 | 0.340 | 0.889 |
| Effective | 61.07 | 67.426 | 0.545 | 0.881 |
| Quality-centred | 61.48 | 63.539 | 0.788 | 0.871 |
| Environment protection-oriented | 60.91 | 70.875 | 0.371 | 0.887 |
| Citizen-oriented | 60.80 | 68.497 | 0.524 | 0.882 |

Source: Authors' computation in SPSS.

Table 3 also reveals the corrected item total correlation, which, with a value superior to 0.4 for each of them, suggests a significant relationship exists between the items of interest.

The same is implemented for the second, third, and fourth groups of items and validated for all 468 observations considered as well. The results are revealed in Table 4 (2), Table 5 (3), respectively Table 6 (4). The second group (2) includes 13 items related to the decisions regarding the social aspects that might have an impact on the responsible management of a smart city. The third group (3) includes 15 items related to the decisions regarding the environmental aspects that might have an impact on the responsible management of a smart city. The fourth group (4) includes 11 items related to the decisions regarding the social aspects that might have an impact on the responsible management of a smart city.

**Table 4.** Reliability and Item-Total Statistics.

| Cronbach's Alpha | Cronbach's Alpha Based on Standardized Items | | | N of Items |
|---|---|---|---|---|
| 0.947 | 0.949 | | | 13 |
| **Item** | **Scale Mean if Item Deleted** | **Scale Variance if Item Deleted** | **Corrected Item-Total Correlation** | **Cronbach's Alpha if Item Deleted** |
| ICT based medical technology | 45.85 | 96.082 | 0.603 | 0.947 |
| Decent dwelling conditions for citizens | 45.74 | 97.330 | 0.733 | 0.943 |
| Non-discriminatory digital social care | 46.13 | 93.152 | 0.800 | 0.941 |

**Table 4.** *Cont.*

| Item | Scale Mean if Item Deleted | Scale Variance if Item Deleted | Corrected Item-Total Correlation | Cronbach's Alpha if Item Deleted |
|---|---|---|---|---|
| Construction and modernisation of smart buildings | 45.81 | 97.664 | 0.727 | 0.943 |
| ICT-based lifelong learning opportunities for citizens | 46.41 | 92.153 | 0.776 | 0.942 |
| Development of vocational schools | 46.33 | 93.665 | 0.733 | 0.943 |
| Digital inclusion of people with disabilities | 45.94 | 96.699 | 0.753 | 0.942 |
| Development of ICT competencies in the educational system | 45.72 | 97.954 | 0.756 | 0.943 |
| Access to digital educational platforms | 45.74 | 97.476 | 0.790 | 0.942 |
| Restoration and modernisation of health spaces | 45.95 | 96.355 | 0.682 | 0.944 |
| Restoration and modernisation of educational spaces | 45.94 | 94.412 | 0.809 | 0.941 |
| Endowment of educational institutions with advanced ICT equipment | 45.78 | 96.015 | 0.793 | 0.941 |
| Telemedical services | 46.26 | 93.026 | 0.750 | 0.943 |

Source: Authors' computation in SPSS.

Table 4 reveals a Cronbach's Alpha coefficient of 0.947 (0.949 for the standardized version), therefore being superior to 0.7 and indicating a significant internal consistency of the said items in the related scale, while the item total statistics confirm the very high contribution of each item to the overall consistency, all items being preserved in the list.

**Table 5.** Reliability and Item-Total Statistics.

| Cronbach's Alpha | Cronbach's Alpha Based on Standardized Items | | N of Items |
|---|---|---|---|
| 0.928 | 0.930 | | 15 |
| **Item** | Scale Mean if Item Deleted | Scale Variance if Item Deleted | Corrected Item-Total Correlation | Cronbach's Alpha if Item Deleted |

| Item | Scale Mean if Item Deleted | Scale Variance if Item Deleted | Corrected Item-Total Correlation | Cronbach's Alpha if Item Deleted |
|---|---|---|---|---|
| Decreasing the air pollution level | 53.63 | 85.873 | 0.432 | 0.929 |
| Decreasing the soil pollution level | 54.15 | 79.107 | 0.826 | 0.919 |
| Decreasing the water pollution level | 53.94 | 81.813 | 0.731 | 0.922 |
| Use of smart equipment for the continuous monitoring of the pollution level | 54.24 | 78.069 | 0.787 | 0.920 |
| Smart eco-automation technology | 54.11 | 81.595 | 0.662 | 0.924 |
| Use of smart equipment for the continuous monitoring of weather | 54.43 | 79.136 | 0.738 | 0.921 |
| Increasing energetic efficiency | 53.95 | 82.141 | 0.667 | 0.924 |

**Table 5.** *Cont.*

| Item | Scale Mean if Item Deleted | Scale Variance if Item Deleted | Corrected Item-Total Correlation | Cronbach's Alpha if Item Deleted |
|---|---|---|---|---|
| Norms and procedures for environmental protection | 54.48 | 75.873 | 0.772 | 0.920 |
| Effective management of disasters | 54.09 | 82.177 | 0.651 | 0.924 |
| Optimum ratio between the number of citizens and the surface of public green spaces | 53.89 | 86.511 | 0.283 | 0.935 |
| Monitoring the proper operation of networks | 54.21 | 78.966 | 0.720 | 0.922 |
| Effective management of natural Resources | 53.87 | 84.356 | 0.601 | 0.925 |
| Smart waste management | 53.64 | 84.660 | 0.664 | 0.925 |
| Access to public facilities | 54.10 | 80.174 | 0.662 | 0.924 |
| Decreasing the phonic pollution level | 54.20 | 78.376 | 0.734 | 0.921 |

Source: Authors' computation in SPSS.

In Table 5, the authors ascertain a Cronbach's Alpha coefficient of 0.928 (0.930 for the standardized version), also superior to 0.7, certifying the internal consistency of the items in the related scale, the visible contribution of each item to the overall consistency being confirmed based on the item total statistics.

**Table 6.** Reliability and Item-Total Statistics.

| Cronbach's Alpha | Cronbach's Alpha Based on Standardized Items | | N of Items |
|---|---|---|---|
| 0.903 | 0.904 | | 11 |
| **Item** | **Scale Mean if Item Deleted** | **Scale Variance if Item Deleted** | **Corrected Item-Total Correlation** | **Cronbach's Alpha if Item Deleted** |
| Stimulating entrepreneurial initiatives | 38.13 | 63.070 | 0.527 | 0.901 |
| Supporting business creation | 38.29 | 62.136 | 0.602 | 0.896 |
| Creating an attractive business environment | 38.38 | 60.879 | 0.697 | 0.891 |
| Digitalization of payments and records | 38.44 | 61.921 | 0.647 | 0.894 |
| Efficient digital use of city resources | 38.62 | 61.945 | 0.643 | 0.894 |
| Development of digital financial ecosystems | 38.62 | 59.366 | 0.730 | 0.889 |
| Development of smart and creative industries | 38.34 | 63.585 | 0.622 | 0.896 |
| Stimulation of e-commerce | 38.99 | 61.981 | 0.546 | 0.900 |
| Creating jobs for minorities and disadvantaged people through new start-ups launching | 38.91 | 58.484 | 0.680 | 0.892 |
| Access to digital platforms providing jobs | 38.64 | 60.916 | 0.672 | 0.893 |
| Online availability of information provided by city authorities for setting up a business | 38.38 | 60.827 | 0.712 | 0.891 |

Source: Authors' computation in SPSS.

As before-mentioned, in Table 6, the authors have a Cronbach's Alpha coefficient superior to 0.7 (0.903, respectively 0.904 for the standardized version), confirming the internal consistency of the items in the related scale and the observed contribution of all items to such consistency.

As Cronbach's Alpha test was validated for all four groups of items, the same are subject to the extraction of the principal components, therefore lowering the number of items contained in each group.

As in the previously rendered section, each group of items is separately treated, with the results revealed in Table 7 (see also Appendix B) (1), Table 8 (see also Appendix C) (2), Table 9 (see also Appendix D) (3), and Table 10 (see also Appendix E).

**Table 7.** Rotated Component Matrix [a].

| Item | Component | | |
|---|---|---|---|
| | 1 | 2 | 3 |
| Visionary | 0.791 | | |
| Proactive | 0.761 | | |
| Ethic | 0.759 | | |
| Transparent | 0.748 | | |
| Equitable | 0.717 | | |
| Participative | 0.692 | | |
| Socially responsible | 0.649 | | |
| Flexible | 0.614 | | |
| Citizen-oriented | 0.597 | | 0.583 |
| Decision-oriented | 0.589 | | |
| Environment protection-oriented | | 0.897 | |
| Effective | | 0.876 | |
| Economically efficient | | 0.867 | |
| Quality-centered | | 0.801 | |
| Innovative | | | 0.876 |
| Open to new | | | 0.839 |

Extraction Method: Principal Component Analysis. Rotation Method: Varimax with Kaiser Normalization.
[a]. Rotation converged in 5 iterations. Source: Authors' computation in SPSS.

The correlation matrix (not provided herein for length-related issues) mainly reveals a statistically significant ($p$-value < 0.01), medium-to-high, positive correlation between items in most of the cases. The correlation matrix determinant, amounting to $1.29 \times 10^{-5}$ and, as a consequence, different from zero, certifies the absence of multi-collinearity.

The KMO and Barlett's test (Appendix A) indicate, by virtue of the value of KMO, amounting to 0.871 (superior to 0.6–0.7), the adequacy of the dataset for being analyzed in terms of principal components, respectively, according to Barlett's test of sphericity, a statistically significant correlation among at least two initial items ($p$-value < 0.01) an aspect certified by the inequality between the correlation matrix and the unitary matrix.

As the eigenvalues, standing for the variances of factors, should exceed 1 for the principal components to be selected (see Appendix B), the authors detected that there are three such components out of the sixteen ones, covering 68.966% of the information incorporated in the initial set of items.

Table 7, revealing the rotated component matrix arising from the rotation of the factorial axes, provides us with all necessary information for the proper interpretation of the newly derived variables, specifically the principal components. The Varimax method

invoked preserves the factors uncorrelated in full; the generated components, not affected by multi-collinearity, are usable as explanatory variables in any regression.

The principal components for the first group of items (1) are in this case, considering their association with the initial items: PC_1.1 (Social and strategic orientation, including the following: Visionary; Proactive; Ethic; Transparent; Equitable; Participative; Socially responsible; Flexible; Citizen-oriented and Decision-oriented); PC_1.2—(Environmental and economic orientation including the following: Environment protection-oriented; Effective; Economically efficient and Quality-centred) and PC_1.3 (Innovativeness, including the following: Innovative and Open to new), defining the potentially necessary characteristics of smart city-centered responsible management.

**Table 8.** Rotated Component Matrix [a].

| Item | Component | |
|---|---|---|
| | 1 | 2 |
| Telemedical services | 0.820 | |
| Endowment of educational institutions with advanced ICT equipment | 0.811 | |
| Development of vocational schools | 0.776 | |
| Restoration and modernisation of educational spaces | 0.776 | |
| ICT-based lifelong learning opportunities for citizens | 0.775 | |
| Restoration and modernisation of health spaces | 0.758 | |
| Access to digital educational platforms | 0.728 | |
| Development of ICT competencies in the educational system | 0.722 | |
| ICT based medical technology | 0.554 | 0.544 |
| Digital inclusion of people with disabilities | | 0.901 |
| Decent dwelling conditions for citizens | | 0.815 |
| Construction and modernisation of smarts building | | 0.681 |
| Non-discriminatory digital social care | 0.551 | 0.650 |

Extraction Method: Principal Component Analysis. Rotation Method: Varimax with Kaiser Normalization.
[a]. Rotation converged in 3 iterations. Source: Authors' computation in SPSS.

The correlation matrix shows a strong, statistically significant (*p*-value < 0.01), positive correlation between items. The determinant of the same, with a value of $8.01 \times 10^{-6}$, indicates the absence of multi-collinearity.

In Appendix C, KMO amounts to 0.916, thus being superior to the acceptance limit of 0.6–0.7 and the data being fit for PCA. Barlett's test of sphericity certifies the significant correlation of at least two initial items (*p*-value < 0.01), the correlation matrix not being unitary.

Two components are kept for further analysis out of the thirteen, as indicated by the eigenvalues displayed in Appendix B.

Table 8 shows the rotated component matrix, revealing, based on the association with the initial items, the principal components for the second group (2), namely: PC_2.1—Health and Education (including the following: Telemedical services; Endowment of educational institutions with advanced ICT equipment; Development of vocational schools; Restoration and modernization of educational spaces; ICT-based lifelong learning opportunities for citizens; Restoration and modernization of health spaces; Access to digital educational platforms; Development of ICT competencies in the educational system and ICT based medical technology) and PC_2.2—Social services and Housing (including the following: Digital inclusion of people with disabilities; Decent dwelling conditions for citizens; Construction and modernisation of smart buildings and Non-discriminatory digital social care), defining the specific social issue-related management decisions potentially beneficial for a smart city.

**Table 9.** Rotated Component Matrix [a].

| Item | Component | |
|---|---|---|
| | 1 | 2 |
| Decreasing the air pollution level | 0.889 | |
| Optimum ratio between the number of citizens and the surface of public green spaces | 0.863 | |
| Smart eco-automation technology | 0.852 | |
| Monitoring the proper operation of networks | 0.828 | |
| Decreasing the soil pollution level | 0.775 | |
| Use of smart equipment for the continuous monitoring of the pollution level | 0.767 | |
| Use of smart equipment for the continuous monitoring of weather | 0.712 | |
| Access to public facilities | 0.637 | |
| Decreasing the phonic pollution level | 0.615 | |
| Decreasing the water pollution level | 0.579 | 0.552 |
| Norms and procedures for environmental protection | | 0.851 |
| Increasing energetic efficiency | | 0.767 |
| Effective management of disasters | | 0.688 |
| Smart waste management | 0.510 | 0.527 |
| Effective management of natural resources | | 0.501 |

Extraction Method: Principal Component Analysis. Rotation Method: Varimax with Kaiser Normalization.
[a]. Rotation converged in 3 iterations. Source: Authors' computation in SPSS.

The correlation matrix reveals a high, statistically significant ($p$-value < 0.01), positive correlation between items. The determinant of the same, with a value of $1.59 \times 10^{-5}$, indicates the absence of multi-collinearity.

In Appendix D, KMO amounts to 0.920, superior to 0.6–0.7, allowing for PCA. Barlett's test of sphericity shows a significant correlation between at least two initial items ($p$-value < 0.01), as the correlation matrix is not unitary.

Two components are kept for further analysis this time too, out of the fifteen ones, as suggested by the eigenvalues displayed in Appendix D.

Table 9, rendering the rotated component matrix, indicates, based on the association with the initial items, the principal components for the third group (3), more exactly: PC_3.1—Anti-pollution actions and environmental monitoring (including the following: Decreasing the air pollution level; Optimum ratio between the number of citizens and the surface of public green spaces; Smart eco-automation technology; Monitoring the proper operation of networks; Decreasing the soil pollution level; Use of smart equipment for the continuous monitoring of the pollution level; Use of smart equipment for the continuous monitoring of weather; Access to public facilities; Decreasing the phonic pollution level and Decreasing the water pollution level) and PC_3.2—Environmental management (including the following: Norms and procedures for environmental protection; Increasing energetic efficiency; Effective management of disasters; Smart waste management; and Effective management of natural resources), defining the environmental management decisions potentially beneficial for a smart city.

The correlation matrix reveals a medium-to-high, statistically significant ($p$-value < 0.01), positive correlation between items.

KMO, rendered in Appendix E, amounts to 0.865, exceeding 0.6–0.7 and, therefore, allowing for the analysis of the principal components. Barlett's test indicates a significant correlation between at least two initial items ($p$-value < 0.01), the correlation matrix not being unitary.

Appendix E also shows two components to be kept for further analysis this time as well, out of the eleven ones, given the related eigenvalues higher than 1.

Table 10 displays the rotated component matrix, with the principal components for the fourth group (4) associated with the initial items: PC_4.1—Digitalization (including the following: Development of digital financial ecosystems; Digitalization of payments and records; Stimulation of e-commerce; Access to digital platforms providing jobs; Development of smart and creative industries; and Efficient digital use of city resources) and PC_4.2—Entrepreneurship (including the following: Supporting business creation; Stimulating entrepreneurial initiatives; Creating an attractive business environment; Online availability of information provided by city authorities for setting up a business; Creating jobs for minorities and disadvantaged people through new start-ups launching); defining the economic aspect-related management decisions potentially beneficial for a smart city.

**Table 10.** Rotated Component Matrix [a].

| Item | Component | |
| --- | :---: | :---: |
| | 1 | 2 |
| Development of digital financial ecosystems | 0.871 | |
| Digitalization of payments and records | 0.861 | |
| Stimulation of e-commerce | 0.822 | |
| Access to digital platforms providing jobs | 0.778 | |
| Development of smart and creative industries | 0.709 | |
| Efficient digital use of city resources | 0.641 | |
| Supporting business creation | | 0.936 |
| Stimulating entrepreneurial initiatives | | 0.928 |
| Creating an attractive business environment | | 0.829 |
| Online availability of information provided by city authorities for setting up a business | | 0.690 |
| Creating jobs for minorities and disadvantaged people through new start-ups launching | 0.522 | 0.544 |

Extraction Method: Principal Component Analysis. Rotation Method: Varimax with Kaiser Normalization.
[a]. Rotation converged in 3 iterations. Source: Authors' computation in SPSS.

Given the initial purpose of our study, namely to determine the relationship between variable (1), designated by the answers to questions Q1.1–Q1.16, reflecting the potentially necessary characteristics of the smart city-centred responsible management (16 items), and variables (2), (3) and (4), rendered via the answers to questions Q2.1–Q2.13, Q3.1–Q3.15, Q4.1–Q4.11, revealing the specific social issue-related management decisions potentially beneficial for a smart city (13 items), the environmental management decisions potentially beneficial for a smart city (15 items), respectively the economic aspect-related management decisions potentially beneficial for a smart city (11 items), and considering the simplification of the procedures by the extraction of the principal components out of such variables, thereby decreasing their volume, the authors decided to test the correlational Spearman and Kendall coefficients (mentioned in the methodological part of the paper), by resorting to the arisen principal components for the four variables of interest (three components for (1) and two components for each remaining variables considered (2), (3) and (4)) as seen in Table 11.

Although the authors tested the correlational relationships between PC_1.1 and PC_2.1, PC_2.2, PC_3.1, PC_3.2, PC_4.1, PC_4.2, between PC_1.2 and PC_2.1, PC_2.2, PC_3.1, PC_3.2, PC_4.1, PC_4.2, respectively between PC_1.3 and PC_2.1, PC_2.2, PC_3.1, PC_3.2, PC_4.1, PC_4.2, only the statistically validated results ($p$-value < 0.01) were rendered in this paper, as reflected by the related tables.

**Table 11.** Correlational analysis Correlation of PC_1.1 with PC_2.2, PC_3.2 and PC_4.2.

| | | | PC_2.2 | PC_3.2 | | PC_4.2 |
|---|---|---|---|---|---|---|
| Kendall's tau_b | PC_1.1 | Correlation Coefficient | 0.371 *** | 0.242 *** | | 0.089 *** |
| | | Sig. (2-tailed) | 0.000 | 0.000 | | 0.004 |
| Spearman's rho | PC_1.1 | Correlation Coefficient | 0.524 *** | 0.353 *** | | 0.135 *** |
| | | Sig. (2-tailed) | 0.000 | 0.000 | | 0.003 |
| | | | PC_3.1 | PC_4.1 | PC_4.2 |
| | | Correlation of PC_1.2 with C_3.1, PC_4.1 and PC_4.2 | | | |
| Kendall's tau_b | PC_1.2 | Correlation Coefficient | 0.168 *** | 0.258 *** | 0.156 *** |
| | | Sig. (2-tailed) | 0.000 | 0.000 | 0.000 |
| Spearman's rho | PC_1.2 | Correlation Coefficient | 0.254 *** | 0.383 *** | 0.232 *** |
| | | Sig. (2-tailed) | 0.000 | 0.000 | 0.000 |
| | | Correlation of PC_1.3 with PC_2.1, PC_3.1 and PC_4.1 | | | |
| | | | PC_2.1 | PC_3.1 | PC_4.1 |
| Kendall's tau_b | PC_1.3 | Correlation Coefficient | 0.434 *** | 0.317 *** | 0.224 *** |
| | | Sig. (2-tailed) | 0.000 | 0.000 | 0.000 |
| Spearman's rho | PC_1.3 | Correlation Coefficient | 0.633 *** | 0.476 *** | 0.326 *** |
| | | Sig. (2-tailed) | 0.000 | 0.000 | 0.000 |

*** Correlation is significant at the 0.01 level (2-tailed). Source: Authors' computation in SPSS.

The authors observe that the principal components of the initial comprehensive variable (1) are correlated at least with one principal component of variables (2), (3), and (4). Save for one correlation per table, outlining a rather low, negative relationship between the variables of interest, the correlations are medium in average and positive, suggesting, overall, that there is a statistically significant connection between the characteristics of smart city-centered responsible management and the specific social/environmental/economic-related management decisions potentially beneficial for the same.

In order to deepen the research, the authors took into consideration the need to detect more than just the simple correlation of variables, looking for the identification of some relationships of dependency between the variables approached in the previous section as explanatory variables and the variables reflected by the answers to five questions Q5.1–Q5.5, measuring the extent to which participants would like to be involved in different social/environmental projects, in volunteering actions focused on social/environmental issues, and in business establishment-related competitions, the latter being taken, each at a time, as explained variable.

Specifically, the authors developed five multiple regressions with a similar structure but coupling items in a logical way.

$$Q5.1 = F(PC\_1.1, PC\_1.2, PC\_1.3, PC\_2.1, PC\_2.2) \tag{1}$$

$$Q5.2 = F(PC\_1.1, PC\_1.2, PC\_1.3, PC\_2.1, PC\_2.2) \tag{2}$$

$$Q5.3 = F(PC\_1.1, PC\_1.2, PC\_1.3, PC\_3.1, PC\_3.2) \tag{3}$$

$$Q5.4 = F(PC\_1.1, PC\_1.2, PC\_1.3, PC\_3.1, PC\_3.2) \tag{4}$$

$$Q5.5 = F(PC\_1.1, PC\_1.2, PC\_1.3, PC\_4.1, PC\_4.2) \tag{5}$$

Equation (1) (Table 12, respectively Appendix F) tests the impact that PC_1.1, PC_1.2 and PC_1.3, representing the potentially necessary characteristics of the smart city-centred responsible management, together with PC_2.1 and PC_2.2, standing for the specific social issue-related management decisions potentially beneficial for a smart city, might exert on Q5.1, precisely on the desire of respondents to get involved in social ICT based projects initiated by the authorities of the city, so as to increase the community life quality, while

Equation (2) (Table 13, respectively Appendix G) checks for the influence exercised by the same regressors on Q5.2, this time, the latter designating the desire of respondents to become volunteers in an association/NGO using ICT, with the aim of solving social community issues.

　　Equation (3) (Table 14, respectively Appendix H) analyses if PC_1.1, PC_1.2 and PC_1.3, reflecting the potentially necessary characteristics of the smart city-centred responsible management, together with PC_3.1 and PC_3.2, revealing the environmental management decisions potentially beneficial for a smart city, might exert influences on Q5.3, representing the desire of respondents to get involved in environmental protection ICT based projects initiated by the authorities of the city, so as to increase the community life quality, while Equation (4) (Table 15, respectively Appendix I) tests the impact of the same regressive variables on Q5.4, the latter describing the desire of respondents to become volunteers in a association/NGO using ICT, with the aim of solving environmental issues.

　　Equation (5) (Table 16 and Appendix J) approaches the possible influence that PC_1.1, PC_1.2, and PC_1.3, representing the potentially necessary characteristics of smart city-centered responsible management, together with PC_4.1 and PC_4.2, standing for the economic aspect-related management decisions potentially beneficial for a smart city, might exert on Q5.5, precisely on the desire of respondents to take part in competitions financed by the authorities of the city so as to set up an ICT-based business.

**Table 12.** Model coefficients [a].

| Model | | Unstandardized Coefficients | | Standardized Coefficients | t | Sig. |
|---|---|---|---|---|---|---|
| | | B | Std. Error | Beta | | |
| | (Constant) | 3.833 | 0.039 | | 97.949 | 0.000 |
| | PC_1.1 | 0.272 | 0.054 | 0.279 | 50.026 | 0.000 |
| 1 | PC_1.2 | −0.014 | 0.040 | −0.015 | −0.357 | 0.721 |
| | PC_1.3 | 0.093 | 0.053 | 0.095 | 10.747 | 0.081 |
| | PC_2.1 | −0.064 | 0.054 | −0.066 | −10.181 | 0.238 |
| | PC_2.2 | 0.268 | 0.054 | 0.275 | 40.993 | 0.000 |

[a]. Dependent Variable: Q5.1. Source: Authors' computation in SPSS.

The output of Equation (1) reveals the R Square (0.256) (see Appendix F), suggesting that the data fit to a certain extent the regression model, or, otherwise said, that 25.6% of the dependent variable variability is explained by the regression model, while the DW test ($2.022 \approx 2$) indicates no autocorrelation of residuals.

　　Appendix F also displays the source of variation, or more precisely, the decomposition of the model total variance into explained/unexplained (residual) variance. The F test, with the associated significance level ($p$-value < 0.01), shows that at least one regression coefficient is different from the null value, therefore indicating the reliability of the present regression model.

　　The regression coefficients, rendered in Table 12, reveal a statistically significant ($p$-value < 0.01) positive influence, beyond the intercept, of PC_1.1 (0.272) and PC_2.2 (0.268), respectively, and a lower statistically significant ($p$-value < 0.1) positive influence of PC_1.3 (0.093) on Q5.1.

　　Given that at least one component for each variable of interest is a statistically significant predictor for the dependent variable, the authors can state that the potentially necessary characteristics of smart city-centered responsible management and the specific social issue-related management decisions potentially beneficial for a smart city exert influences on the desire of respondents to get involved in social ICT-based projects initiated by the authorities of the city so as to increase the quality of community life.

**Table 13.** Regression coefficients [a].

| Model | | Unstandardized Coefficients | | Standardized Coefficients | t | Sig. |
|---|---|---|---|---|---|---|
| | | B | Std. Error | Beta | | |
| | (Constant) | 3.359 | 0.042 | | 80.674 | 0.000 |
| | PC_1.1 | 0.068 | 0.058 | 0.064 | 1.177 | 0.240 |
| | PC_1.2 | 0.110 | 0.043 | 0.104 | 2.576 | 0.010 |
| 2 | PC_1.3 | 0.233 | 0.057 | 0.222 | 4.112 | 0.000 |
| | PC_2.1 | 0.329 | 0.058 | 0.313 | 5.688 | 0.000 |
| | PC_2.2 | 0.125 | 0.057 | 0.118 | 2.182 | 0.030 |

[a]. Dependent variable: Q5.2. Source: Authors' computation in SPSS.

Equation (2) indicates, via the R Square (0.276) (see Appendix G), that 27.6% of the dependent variable variability is explained by the regression model, with the DW test (2.104) outlining the absence of autocorrelation of errors.

The statistically significant ($p$-value < 0.01) F test (see Appendix G) supports the representativeness of the related regression model.

The regression coefficients, rendered in Table 13, reveal a statistically significant ($p$-value < 0.01) positive influence of all items on the dependent variable, save for PC_1.1, suggesting that the potentially necessary characteristics of smart city-centered responsible management and the specific social issue-related management decisions potentially beneficial for a smart city exert influences on the desire of respondents to become volunteers in an association/NGO using ICT with the aim of solving social community issues.

**Table 14.** Regression coefficients [a].

| Model | | Unstandardized Coefficients | | Standardized Coefficients | t | Sig. |
|---|---|---|---|---|---|---|
| | | B | Std. Error | Beta | | |
| | (Constant) | 3.662 | 0.040 | | 90.917 | 0.000 |
| | PC_1.1 | 0.167 | 0.044 | 0.174 | 3.807 | 0.000 |
| | PC_1.2 | 0.031 | 0.041 | 0.032 | 0.750 | 0.454 |
| 3 | PC_1.3 | 0.097 | 0.049 | 0.101 | 1.979 | 0.048 |
| | PC_3.1 | 0.322 | 0.048 | 0.336 | 6.685 | 0.000 |
| | PC_3.2 | 0.050 | 0.045 | 0.052 | 1.107 | 0.269 |

[a]. Dependent variable: Q5.3. Source: Authors' computation in SPSS.

The output of Equation (3) reveals a lower R Square (0.185) (see Appendix H), therefore only 18.5% of the variability of the dependent variable is explained by the regression model; the DW test (2.105) suggests, however, the absence of autocorrelation of errors.

With the statistically significant ($p$-value < 0.01) F test (see Appendix H), the output stands for the adequacy of the regression model.

The regression coefficients, rendered in Table 14, show a statistically significant ($p$-value < 0.01) positive influence of the dependent variable, save for the intercept, of PC_1.1 (0.167) and PC_3.1 (0.322), PC_1.3 (0.097) being in exchange significant for a $p$-value < 0.05, confirming, given that at least one components of each considered variable is an appropriate regressor for the explained variable, that the potentially necessary characteristics of the smart city-centred responsible management and the environmental management decisions potentially beneficial for a smart city exert influences on the desire of respondents to get involved in environmental protection ICT based projects initiated by the authorities of the city, so as to increase the community life quality.

**Table 15.** Regression coefficients [a].

| Model | | Unstandardized Coefficients | | Standardized Coefficients | T | Sig. |
|---|---|---|---|---|---|---|
| | | B | Std. Error | Beta | | |
| | (Constant) | 3.432 | 0.037 | | 93.112 | 0.000 |
| | PC_1.1 | 0.119 | 0.040 | 0.111 | 2.971 | 0.003 |
| | PC_1.2 | 0.112 | 0.037 | 0.104 | 2.976 | 0.003 |
| 4 | PC_1.3 | 0.242 | 0.045 | 0.225 | 5.409 | 0.000 |
| | PC_3.1 | 0.554 | 0.044 | 0.515 | 12.563 | 0.000 |
| | PC_3.2 | 0.015 | 0.041 | 0.014 | 0.361 | 0.718 |

[a]. Dependent variable: Q5.4. Source: Authors' computation in SPSS.

The implementation of Equation (4) reveals a medium to high level of R Square (0.457) (see Appendix I); therefore, 45.7% of the variability of the dependent variable is explained by the regression model, while the DW test (2.147) stands for the non-autocorrelation of errors.

The F test, encountered in Appendix I, being statistically significant ($p$-value $< 0.01$), reveals the adequacy of the regression model.

The regression coefficients, displayed in Table 15, have a statistically significant ($p$-value $< 0.01$) positive influence on the dependent variable, save for PC_3.2, allowing us to confirm that the potentially necessary characteristics of smart city-centered responsible management and the environmental management decisions potentially beneficial for a smart city exert influences on the desire of respondents to become volunteers in an association/NGO using ICT with the aim of solving environmental issues.

**Table 16.** Regression coefficients [a].

| Model | | Unstandardized Coefficients | | Standardized Coefficients | t | Sig. |
|---|---|---|---|---|---|---|
| | | B | Std. Error | Beta | | |
| | (Constant) | 4.107 | 0.034 | | 119.477 | 0.000 |
| | PC_1.1 | 0.047 | 0.036 | 0.053 | 1.299 | 0.195 |
| | PC_1.2 | 0.049 | 0.040 | 0.055 | 1.225 | 0.221 |
| 5 | PC_1.3 | 0.032 | 0.039 | 0.036 | 0.829 | 0.408 |
| | PC_4.1 | 0.111 | 0.040 | 0.124 | 2.745 | 0.006 |
| | PC_4.2 | 0.464 | 0.040 | 0.519 | 11.512 | 0.000 |

[a]. Dependent variable: Q5.5. Source: Authors' computation in SPSS.

The output generated as a result of the implementation of Equation (5) indicates a medium level of R Square (0.314) (see Appendix J), showing that 31.4% of the variability of the dependent variable is explained by the regression model, while the DW test (2.080) reveals the absence of autocorrelation of residuals.

Appendix J also renders the decomposition of the model total variance into explained/unexplained variance and the statistically significant ($p$-value $< 0.01$) F test, outlining the model reliability.

The regression coefficients, displayed in Table 16, show a statistically significant ($p$-value $< 0.01$) positive influence of the intercept, PC_4.1, and PC_4.2 on the dependent variable, suggesting that only the economic aspect-related management decisions potentially beneficial for a smart city exert influences on the desire of respondents to take part in competitions financed by the authorities of the city so as to set up an ICT-based business.

## 5. Discussion

Despite the fact that responsible decision-making in the context of smart cities represents a research topic frequently approached by authors in empirical studies, a relatively small number of these papers are focused on exploring and understanding the perceptions and attitudes of Gen Z students towards the responsible management of smart cities. Our research outcomes revealed that Romanian final-year undergraduate students perceive the characteristics of smart city-centered responsible management (grouped in the following three categories: social and strategic orientation; environmental and economic orientation; and innovativeness) as being strongly connected to specific social, environmental, and economic-related management decisions and vice versa.

For instance, the first hypothesis (H1) states that the social and strategic characteristics of smart city-centered responsible management and the managerial decisions regarding social issue solving by initiating social service and housing actions are interdependent. However, the statistical analysis highlights that the principal components of the social and strategic orientation of smart city-centered responsible management are correlated at least with one principal component of social services and housing-related management decisions. Similar to other studies, our research highlights that Romanian final-year undergraduate students perceive digital inclusion of people with disabilities [88–90] as well as ensuring decent dwelling conditions for citizens [91,92] as results of a proactive, equitable, socially responsible, and citizen-oriented management approach. Additionally, the general development of smart city services speeds up management decisions that, in turn, can enable a social orientation as it also improves the provision of information services to citizens and businesses and solves other problems such as comfortable living conditions that are part of housing options [93]. Ensuring decent dwelling conditions and the construction and modernization of smart buildings can be part of the new urbanism and smart growth principles that are taken into account in the city's planning, as they deal with quality architecture and urban design [94] with action plans such as providing comfortable housing [95] options [96].

In contrast, by analyzing existing literature regarding responsible management in the context of smart cities, the authors found no evidence to support the idea that non-discriminatory digital social care is related, to some extent, to the social and strategic characteristics of smart city-centered responsible management. Consequently, the first hypothesis was partially validated.

The second hypothesis (H2) underlines that the social and strategic characteristics of smart city-centered responsible management and the managerial decisions regarding environmental issue solving by initiating environmental management actions are interdependent. Nevertheless, the statistical analysis suggests that the principal components of the social and strategic orientation of smart city-centered responsible management are correlated at least with one principal component of environmental management decisions. Consistent with the results of other studies, our research outcomes indicate that Romanian final-year undergraduate students perceive environmental management decisions such as developing norms and establishing procedures for environmental protection [97], as well as increasing energetic efficiency [98], smart waste management [99], effective management of disasters [100], and of natural resources [101], as results of a visionary management approach. In contrast, by analyzing existing literature regarding responsible management in the context of smart cities, the authors found no evidence to support the idea that environmental management decisions are related, to some extent, to the proactive, ethical, transparent, equitable, participative, socially responsible, flexible, citizen-oriented, and decision-oriented characteristics of smart city-centered responsible management. Hence, the second hypothesis was partially validated.

The third hypothesis (H3) states that the social and strategic characteristics of smart city-centered responsible management and the managerial decisions regarding the stimulation and support of entrepreneurial initiatives are interdependent. Nevertheless, the statistical analysis indicates that the principal components of the social and strategic ori-

entation of smart city-centered responsible management are correlated at least with one principal component of entrepreneurship-related management decisions. Similar to other studies, our research illustrates that Romanian final-year undergraduate students perceive entrepreneurship management decisions such as creating jobs for minorities and disadvantaged people through new start-ups launching opportunities as the result of a proactive and socially responsible management approach [102]. Moreover, entrepreneurship management decisions such as supporting business creation and providing online available information for setting up a business are perceived by Romanian final-year undergraduate students as results of a participative management approach, and in addition, these results are congruent with those of other studies [103–105]. In contrast, by analyzing existing literature regarding responsible management in the context of smart cities, the authors found no evidence to support the idea that entrepreneurial managerial decisions such as stimulating entrepreneurial initiatives and creating an attractive business environment are related, to some extent, to the social and strategic characteristics of smart city-centered responsible management. However, other studies suggest that stimulating entrepreneurial initiatives and creating an attractive business environment are the results of an economic-oriented management approach [105]. Therefore, the third hypothesis was partially validated.

The fourth hypothesis (H4) underlines that the environmental and economic characteristics of smart city-centered responsible management and the managerial decisions regarding environmental issue solving by initiating anti-pollution and environmental monitoring actions are interdependent. However, the statistical analysis suggests that the principal components of the environmental and economic orientation of smart city-centered responsible management are correlated at least with one principal component of anti-pollution and environmental monitoring-related management decisions. Similar to other studies, our research highlights that Romanian final-year undergraduate students perceive anti-pollution and environmental monitoring management decisions such as the use of smart equipment for the continuous monitoring of the pollution level [106], decreasing the air pollution level [107], and the water pollution level [108] as results of an environment protection-oriented and effective management approach. Additionally, a strong case is to be made in reference to Singapore's governmental approach in terms of environmental pollution; one action is the decision to invest in transit road infrastructure and encourage people to use public transport as a solution to reduce pollution [109]. Another study [110] shows that the lack of environmental orientation in city management is a barrier to creating a smart city.

In contrast, by analyzing existing literature regarding responsible management in the context of smart cities, the authors found no evidence to support the idea that anti-pollution and environmental monitoring management decisions such as decreasing the soil pollution level, monitoring the proper operation of networks, using smart equipment for the continuous monitoring of weather, accessing public facilities, decreasing the phonic pollution level, and ensuring an optimum ratio between the number of citizens and the surface of public green spaces are related, to some extent, to the environmental and economic characteristics of smart city-centered responsible management. Consequently, the fourth hypothesis has been partially validated.

The fifth hypothesis (H5) states that the environmental and economic characteristics of smart city-centered responsible management and the managerial decisions regarding the digitalization of smart cities are interdependent. Nevertheless, the statistical analysis illustrates that the principal components of the environmental and economic orientation of smart city-centered responsible management are correlated at least with one principal component of digitalization-related management decisions. Similar to other studies, our research outcomes indicate that Romanian final-year undergraduate students perceive digitalization-related management decisions such as stimulation of e-commerce [111] and digitalization of payments and records [112] as results of an effective management approach. In contrast, by analyzing existing literature regarding responsible management in the context of smart cities, the authors found no evidence to support the idea that

digitalization-related management decisions such as the development of digital financial ecosystems, ensuring citizens' access to digital platforms providing jobs, the development of smart and creative industries, and efficient digital use of city resources are related, to some extent, to the environmental and economic characteristics of smart city-centered responsible management. Consequently, the fifth hypothesis was partially validated.

The sixth hypothesis (H6) illustrates that the environmental and economic characteristics of smart city-centered responsible management and entrepreneurship-related management decisions are interdependent. However, the statistical analysis highlights that the principal components of the environmental and economic orientation of smart city-centered responsible management are correlated at least with one principal component of entrepreneurship-related management decisions. Consistent with the results of other studies, our research outcomes indicate that Romanian final-year undergraduate students perceive entrepreneurship-related management decisions, such as stimulating entrepreneurial initiatives, as a result of an economically efficient management approach [113]. In contrast, by analyzing existing literature regarding responsible management in the context of smart cities, the authors found no evidence to support the idea that entrepreneurship-related management decisions such as supporting business creation, creating an attractive business environment, providing online availability of information for setting up a business, and creating jobs for minorities and disadvantaged people through the launch of new start-ups are related, to some extent, to the environmental and economic characteristics of smart city-centered responsible management. In consequence, the sixth hypothesis was partially validated.

The seventh hypothesis (H7) states that the innovative characteristics of smart city-centered responsible management and the health and education-related management decisions are interdependent. Nevertheless, the statistical analysis highlights that the principal components of the innovativeness orientation of smart city-centered responsible management are correlated at least with one principal component of health and education-related management decisions. Similar to other studies, our research highlights that Romanian final-year undergraduate students perceive health and education-related management decisions such as providing telemedical services to citizens [114] and access to digital educational platforms [115] as results of an openness to a new management approach. Additionally, one study [116] has shown a direct implication between the municipal administration of the city and its ability to promote innovation and attract both private companies and universities (the educational component) to become municipal government partners.

In contrast, by analyzing existing literature regarding responsible management in the context of smart cities, the authors found no evidence to support the idea that health and education-related management decisions such as the endowment of educational institutions with advanced ICT equipment, the development of vocational schools, providing ICT-based lifelong learning opportunities, the development of ICT competencies in the educational system, and restoring and modernizing educational and health spaces are related, to some extent, to the innovativeness characteristics of smart city-centered responsible management. Consequently, the seventh hypothesis was partially validated.

The eight hypothesis (H8) highlights that the innovative characteristics of smart city-centered responsible management and the managerial decisions regarding environmental issue solving by initiating anti-pollution and environmental monitoring actions are interdependent. However, the statistical analysis illustrates that the principal components of the innovativeness orientation of smart city-centered responsible management are correlated at least with one principal component of anti-pollution and environmental monitoring-related management decisions. In line with the outcomes of other studies, our research results indicate that Romanian final-year undergraduate students perceive anti-pollution and environmental monitoring-related management decisions such as smart eco-automation technology [117] as a result of an innovative management approach.

In contrast, by analyzing existing literature regarding responsible management in the context of smart city, no evidence was found by the authors to support the idea that anti-pollution and environmental monitoring-related management decisions such as decreasing the air pollution level, providing and optimum ratio between the number of citizens and the surface of public green spaces, monitoring the proper operation of networks, decreasing the soil pollution level, the use of smart equipment for the continuous monitoring of the pollution level, the use of smart equipment for the continuous monitoring of weather, providing citizens' with access to public facilities, decreasing the phonic pollution level and decreasing of the water pollution level are related, to some extent, to the innovativeness characteristics of the smart city-centred responsible management. In consequence, the eight hypothesis were partially validated.

The ninth hypothesis (H9) states that the innovative characteristics of smart city-centered responsible management and the managerial decisions regarding the digitalization of smart cities are interdependent. Nevertheless, the statistical analysis indicates that the principal components of the innovativeness orientation of smart city-centered responsible management are correlated at least with one principal component of digitalization-related management decisions. Similar to other studies, our research highlights that the Romanian final-year undergraduate students perceive digitalization-related management decisions, such as the development of smart and creative industries [118], as a result of an innovative and open-minded management approach. Additionally, the connection can be seen in a Smart City project [119] that creates digital platforms to even co-interest citizens into solving city problems and in Helsinki, one attractive city for agile smart city experiments that foster digitalization and the emergence of data-based innovations, mentioning that a smart city is also related to the manner in which it governs its ICT data and systems [120].

In contrast, by analyzing existing literature regarding responsible management in the context of smart cities, the authors found no evidence to support the idea that digitalization-related management decisions such as the development of digital financial ecosystems, the stimulation of e-commerce, ensuring citizens' access to digital platforms providing jobs, the efficient digital use of city resources, and the digitalization of payments and records are related, to some extent, to the innovativeness characteristics of smart city-centered responsible management. Consequently, the ninth hypothesis was partially validated.

The tenth hypothesis (H10) presents the positive influence of the characteristics of the smart city and the specific social management decisions on Gen Z students' desire to get involved in social ICT-based projects initiated by the authorities of the city and NGOs. Concerning the students' willingness to be involved in social projects initiated by the city authorities that involve the use of ICT to increase the quality of life of the community, out of the 468 respondents, 118 ($\approx$25.2%) totally/fully agree, 217 ($\approx$46.4%) agree, 79 ($\approx$16.9%) neither agree nor disagree, 45 ($\approx$9.6%) disagree, and 9 ($\approx$1.9%) completely/fully disagree.

In connection with their desire to become volunteers for an association/NGO that uses ICT to solve the community's social problems, out of the 468 respondents, 70 ($\approx$15%) totally/fully agree, 143 ($\approx$30.5%) agree, 160 ($\approx$34.2%) neither agree nor disagree, 75 ($\approx$16%) disagree, and 20 ($\approx$4.3%) completely/fully disagree.

The results validate the hypothesis, highlighting Gen Z's general increased interest in being involved in various social endeavors.

The eleventh hypothesis (H11) presents the positive influence of the characteristics of the smart city and the specific environmental management decisions on Gen Z students' desire to get involved in social ICT-based projects initiated by the authorities of the city and NGOs. Regarding the respondents' readiness to be involved in environmental protection projects initiated by city authorities and involving the use of ICT to increase the community's quality of life, out of 468 respondents, 95 ($\approx$20.3%) totally/fully agree, 181 ($\approx$38.7%) agree, 138 ($\approx$29.5%) neither agree nor disagree, 47 ($\approx$10%) disagree, and 7 ($\approx$1.5%) completely/fully disagree.

Regarding the respondents' disposition to become volunteers for an association/NGO that uses ICT to solve environmental problems, out of 468 respondents, 85 ($\approx$18.2%)

totally/fully agree, 147 ($\approx$31.4%) agree, 134 ($\approx$28.6%) neither agree nor disagree, 89 ($\approx$19%) disagree, and 13 ($\approx$2.8%) completely/fully disagree.

The results validate the hypothesis, supporting Gen Z's general willingness to be involved in various projects, such as environmental ones.

The twelfth hypothesis (H12) presents the positive influence of the characteristics of the smart city-centered economy and the specific economic management decisions on the Gen Z students' desire to get involved in economic ICT-based projects initiated by the authorities of the city and NGOs. With regard to the students' preparedness to participate in competitions funded by city authorities for starting a business involving the use of ICT, out of the 468 respondents, 168 ($\approx$35.9%) totally/fully agree, 220 ($\approx$47%) agree, 50 ($\approx$10.7%) neither agree nor disagree, 22 ($\approx$4.7%) disagree, and 8 ($\approx$1.7%) completely/fully disagree.

The results strongly reveal Gen Z students' general readiness to be involved in various projects, such as economic-related ones, especially since the respondents have studied at the Administration and Business Faculty. The hypothesis is partially validated because there was not a significant correlation between their involvement in NGOs entrepreneurial initiatives.

## 6. Conclusions

In spite of its relative novelty, the concept of a smart city has become a topic of interest for both theoreticians and practitioners in the post-modern era. Moreover, it involves the implementation of a new type of management entitled "responsible management." Responsible management should take into account not only the stakeholders of a smart city but also its social, economic, and environmental issues. Being highly connected with ICT, responsible management constitutes a major concern for all citizens of a smart city, especially for Gen Z representatives, such as students.

From a theoretical perspective, the study fills an existing research gap regarding the relationships between the concepts of smart cities, responsible management, and Gen Z students. It provides new insights regarding the students' perceptions and attitudes towards the responsible management of a smart city. In this respect, the results show that, according to Gen Z students, the responsible management of smart cities should be driven by several key vectors, such as social responsibility and economic efficiency. Also, the paper identifies several blocks of variables (e.g., the social and strategic orientation of the responsible management) that are correlated with specific managerial decisions of city governments (e.g., social services and housing).

From a practical point of view, the results of this study show that the responsible management of smart cities should identify and analyze the perceptions of Gen Z students and address their needs. Also, the outcomes of this study demonstrate that students are aware of the role played by the city government in ensuring responsible management of the economic, social, and environmental issues of a smart city. Also, they emphasize that most of them are involved in or would like to be involved in different projects designed to expand the use of the principles of smart cities in their place of residence. These results may represent the starting point for building an open platform among the young people of Generation Z to identify and propose new and innovative solutions for city governments that are beneficial to improving the quality of life of the residents. Moreover, smart city-centered responsible management should carry out specific activities aimed at satisfying some key needs, such as the inclusion of people with disabilities, the development of norms and procedures for environmental protection, and the creation of an attractive business environment.

The results of this paper should be interpreted in light of some limitations. First, the research analyzes a limited number of items and variables that influence Gen Z students' perceptions and attitudes. Second, the sample cannot be considered fully representative of the entire Gen Z population due to its size and structure. The questionnaire was applied only to students from two specializations (Business and Administration), belonging to one public higher education institution. Third, the answers provided by the students may be subject to bias, knowing that the authors are teaching at the same faculty. Fourth, the

concept of a smart city and its responsible management represent relatively new topics in both the academic environment and practice.

Further research may take into consideration a larger number of items specific to each of the four variables and/or other variables in order to analyze their impact on Gen Z students' perceptions and attitudes. Moreover, they can lead to new correlations among the variables and/or their items. Also, further research may address a larger sample of Gen Z students from other specializations and other higher education institutions located in the same country or other countries. Also, qualitative research may be beneficial in order to better understand the perceptions and attitudes of Gen Z students towards the responsible management of smart cities.

The originality of the paper is twofold. First, it provides a theoretical model based on the identification of several variables and their specific items related to the dimensions of smart cities. Second, it analyzes their influence on Gen Z students' perceptions and attitudes towards the responsible management of smart cities.

**Author Contributions:** Conceptualization, S.-G.T., C.G., and A.M.; methodology, S.-G.T.; software, O.-S.H.; validation, S.-G.T., C.G., A.M., and O.-S.H.; formal analysis, O.-S.H.; investigation, S.-G.T., C.G., A.M.; resources, S.-G.T., C.G., and A.M.; data curation, O.-S.H.; writing—original draft preparation, S.-G.T., C.G., O.-S.H., and A.M.; writing—review and editing, S.-G.T., C.G., O.-S.H., and A.M.; visualization, S.-G.T.; supervision, S.-G.T.; project administration, S.-G.T. All authors have read and agreed to the published version of the manuscript.

**Funding:** This research received no external funding.

**Institutional Review Board Statement:** Not applicable.

**Informed Consent Statement:** Informed consent was obtained from all subjects involved in the study.

**Data Availability Statement:** The data presented in this study are available on request.

**Conflicts of Interest:** The authors declare no conflict of interest.

## Appendix A

## Perceptions and Attitudes of Generation Z Students towards Responsible Management of a Smart City

The respondent's consent to the GDPR policy is implied by voluntarily completing this questionnaire.
The purpose of the research is strictly scientific, and the results will be published in specialist journals.

* Indicates required question

1. According to your year of birth, you are a representative of: *
   *Mark only one oval.*

   ◯ The Silent Generation (1928-1945)

   ◯ The Baby-Boomers Generation (1946-1964)

   ◯ The Generation X (1965-1980)

   ◯ The Generation Y (1981-1996)

   ◯ The Generation Z (1997-2012)

   ◯ The Generation Alpha (2013-present)

2. Are you familiar with the concept of smart city? *
   *Mark only one oval.*

   ◯ Yes

   ◯ No

3. To what extent do you consider that the responsible management of a smart city needs to be... *
   *Mark only one oval.*

| | Not Important At All | Slightly Important | Neither Unimportant or Important | Important | Very Important |
|---|---|---|---|---|---|
| ...Transparent? | ◯ | ◯ | ◯ | ◯ | ◯ |
| ...Participative? | ◯ | ◯ | ◯ | ◯ | ◯ |
| ...Proactive? | ◯ | ◯ | ◯ | ◯ | ◯ |
| ...Socially responsible? | ◯ | ◯ | ◯ | ◯ | ◯ |
| ...Open to new? | ◯ | ◯ | ◯ | ◯ | ◯ |
| ...Innovative? | ◯ | ◯ | ◯ | ◯ | ◯ |
| ...Flexible? | ◯ | ◯ | ◯ | ◯ | ◯ |
| ...Equitable? | ◯ | ◯ | ◯ | ◯ | ◯ |
| ...Ethic? | ◯ | ◯ | ◯ | ◯ | ◯ |
| ...Visionary? | ◯ | ◯ | ◯ | ◯ | ◯ |
| ...Decision-oriented? | ◯ | ◯ | ◯ | ◯ | ◯ |
| ...Economically efficient? | ◯ | ◯ | ◯ | ◯ | ◯ |
| ...Effective? | ◯ | ◯ | ◯ | ◯ | ◯ |
| ...Quality-centred? | ◯ | ◯ | ◯ | ◯ | ◯ |
| ...Environment protection-oriented? | ◯ | ◯ | ◯ | ◯ | ◯ |
| ...Citizen-oriented? | ◯ | ◯ | ◯ | ◯ | ◯ |

4. To what extent do you consider that decisions regarding the following social aspects have an impact on the responsible management of a smart city? *
*Mark only one oval.*

|  | Strongly disagree | Disagree | Neither agree nor disagree | Agree | Strongly agree |
|---|---|---|---|---|---|
| ICT based medical technology | ○ | ○ | ○ | ○ | ○ |
| Decent dwelling conditions for citizens | ○ | ○ | ○ | ○ | ○ |
| Non-discriminatory digital social care | ○ | ○ | ○ | ○ | ○ |
| Construction and modernization of smart buildings | ○ | ○ | ○ | ○ | ○ |
| ICT-based lifelong learning opportunities for citizens | ○ | ○ | ○ | ○ | ○ |
| Development of vocational schools | ○ | ○ | ○ | ○ | ○ |
| Digital inclusion of people with disabilities | ○ | ○ | ○ | ○ | ○ |
| Development of ICT competencies in the educational system | ○ | ○ | ○ | ○ | ○ |
| Access to digital educational platforms | ○ | ○ | ○ | ○ | ○ |
| Restoration and modernization of health spaces | ○ | ○ | ○ | ○ | ○ |
| Restoration and modernization of educational spaces | ○ | ○ | ○ | ○ | ○ |
| Endowment of educational institutions with advanced ICT equipment | ○ | ○ | ○ | ○ | ○ |
| Telemedical services | ○ | ○ | ○ | ○ | ○ |

5. To what extent do you consider that decisions regarding the following environmental aspects have an impact on the responsible management of a smart city? *
*Mark only one oval.*

|  | Strongly disagree | Disagree | Neither agree nor disagree | Agree | Strongly agree |
|---|---|---|---|---|---|
| Decreasing the air pollution level | ○ | ○ | ○ | ○ | ○ |
| Optimum ratio between the number of citizens and the surface of public green spaces | ○ | ○ | ○ | ○ | ○ |
| Smart eco-automation technology | ○ | ○ | ○ | ○ | ○ |
| Monitoring the proper operation of networks | ○ | ○ | ○ | ○ | ○ |
| Decreasing the soil pollution level | ○ | ○ | ○ | ○ | ○ |
| Use of smart equipment for the continuous monitoring of the pollution level | ○ | ○ | ○ | ○ | ○ |
| Use of smart equipment for the continuous monitoring of weather | ○ | ○ | ○ | ○ | ○ |
| Access to public facilities | ○ | ○ | ○ | ○ | ○ |
| Decreasing the phonic pollution level | ○ | ○ | ○ | ○ | ○ |
| Decreasing the water pollution level | ○ | ○ | ○ | ○ | ○ |
| Norms and procedures for environmental protection | ○ | ○ | ○ | ○ | ○ |
| Increasing energetic efficiency | ○ | ○ | ○ | ○ | ○ |
| Effective management of disasters | ○ | ○ | ○ | ○ | ○ |
| Smart waste management | ○ | ○ | ○ | ○ | ○ |
| Effective management of natural resources | ○ | ○ | ○ | ○ | ○ |

6.  To what extent do you consider that decisions regarding the following economic aspects have an impact on the responsible management of a smart city? *

    *Mark only one oval.*

| | Strongly disagree | Disagree | Neither agree nor disagree | Agree | Strongly agree |
|---|---|---|---|---|---|
| Stimulating entrepreneurial initiatives | ◯ | ◯ | ◯ | ◯ | ◯ |
| Supporting business creation | ◯ | ◯ | ◯ | ◯ | ◯ |
| Creating an attractive business environment | ◯ | ◯ | ◯ | ◯ | ◯ |
| Digitalization of payments and records | ◯ | ◯ | ◯ | ◯ | ◯ |
| Efficient digital use of city resources | ◯ | ◯ | ◯ | ◯ | ◯ |
| Development of digital financial ecosystems | ◯ | ◯ | ◯ | ◯ | ◯ |
| Development of smart and creative industries | ◯ | ◯ | ◯ | ◯ | ◯ |
| Stimulation of e-commerce | ◯ | ◯ | ◯ | ◯ | ◯ |
| Creating jobs for minorities and disadvantaged people through new start-ups launching | ◯ | ◯ | ◯ | ◯ | ◯ |
| Access to digital platforms providing jobs | ◯ | ◯ | ◯ | ◯ | ◯ |
| Online availability of information provided by city authorities for setting up a business | ◯ | ◯ | ◯ | ◯ | ◯ |

### Smart cities- Social, environmental, and economic projects

7.  Have you been involved in projects that focused on the following aspects? *

    *Mark only one oval.*

| | Yes | No |
|---|---|---|
| Social | ☐ | ☐ |
| Environmental | ☐ | ☐ |
| Economic | ☐ | ☐ |

8. To what extent do you agree with the following statements: *
   *Mark only one oval.*

|  | Strongly disagree | Disagree | Neither agree nor disagree | Agree | Strongly agree |
|---|---|---|---|---|---|
| I would like to get involved in social projects initiated by the city authorities and which involve the use of ICT to increase the quality of life of the community. | ⬭ | ⬭ | ⬭ | ⬭ | ⬭ |
| I would like to become a volunteer for an association/NGO that uses ICT to solve social problems of the community. | ⬭ | ⬭ | ⬭ | ⬭ | ⬭ |
| I would like to get involved in environmental protection projects initiated by the city authorities and which involve the use of ICT to increase the quality of life of the community. | ⬭ | ⬭ | ⬭ | ⬭ | ⬭ |
| I would like to become a volunteer for an association/NGO that uses ICT to solve environmental problems. | ⬭ | ⬭ | ⬭ | ⬭ | ⬭ |
| I would like to participate in competitions funded by city authorities for starting a business involving the use of ICT. | ⬭ | ⬭ | ⬭ | ⬭ | ⬭ |

9. Your gender is: *
   *Mark only one oval.*

   ( ) Male

   ( ) Female

   ( ) Non-binary

10. Your current employment status is: *
    *Mark only one oval.*

    ( ) Worker without a contract

    ( ) Full-time employee

    ( ) Part-time employee

    ( ) Business owner

    ( ) Self-employed individual with a registered business

    ( ) Not employed at all

11. What is your field of activity? *
*Mark only one oval.*

○ Not the case

○ Industry

○ Trade

○ Tourism

○ Finance/Banking

○ Education

○ Agriculture

○ Recreation, leisure or sports

○ ICT

○ Other

12. What field of study did you pursue in high school? *
*Mark only one oval.*

○ Theoretical

○ Aptitude-based (Vocational/Art/Sports/Theological)

○ Technological

○ Economic

○ Other

13. What field of study are you currently pursuing in the Faculty of Administration and Business?
*Mark only one oval.*

○ Business Administration (in Romanian)

○ Marketing

○ Business Administration (in English)

○ Public Administration

○ Economic Cybernetics

14. I was born in: *
*Mark only one oval.*

○ A rural region

○ An urban region

15. Your current place of residence is situated in: *
*Mark only one oval.*

○ A rural region

○ An urban region

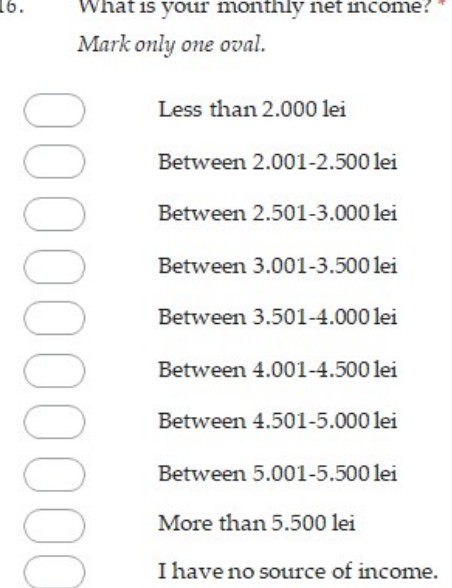

16.    What is your monthly net income? *

*Mark only one oval.*

◯ Less than 2.000 lei

◯ Between 2.001-2.500 lei

◯ Between 2.501-3.000 lei

◯ Between 3.001-3.500 lei

◯ Between 3.501-4.000 lei

◯ Between 4.001-4.500 lei

◯ Between 4.501-5.000 lei

◯ Between 5.001-5.500 lei

◯ More than 5.500 lei

◯ I have no source of income.

## Appendix B

**Table A1.** KMO and Bartlett's Test.

| | | |
|---|---|---|
| Kaiser-Meyer-Olkin Measure of Sampling Adequacy. | | 0.871 |
| Bartlett's Test of Sphericity | Approx. Chi-Square | 5189.112 |
| | df | 120 |
| | Sig. | 0.000 |

Source: Authors' computation in SPSS.

**Table A2.** Total Variance Explained [a].

| Component | Initial Eigenvalues | | | Extraction Sums of Squared Loadings | | | Rotation Sums of Squared Loadings | | |
|---|---|---|---|---|---|---|---|---|---|
| | Total | % of Variance | Cumulative % | Total | % of Variance | Cumulative % | Total | % of Variance | Cumulative % |
| 1 | 6.337 | 39.604 | 39.604 | 6.337 | 39.604 | 39.604 | 5.036 | 31.477 | 31.477 |
| 2 | 3.425 | 21.404 | 61.008 | 3.425 | 21.404 | 61.008 | 3.666 | 22.915 | 54.392 |
| 3 | 1.278 | 7.988 | 68.996 | 1.278 | 7.988 | 68.996 | 2.337 | 14.604 | 68.996 |
| 4 | 0.978 | 6.109 | 75.105 | | | | | | |
| 5 | 0.721 | 4.508 | 79.613 | | | | | | |
| 6 | 0.562 | 3.513 | 83.126 | | | | | | |
| 7 | 0.463 | 2.895 | 86.021 | | | | | | |
| 8 | 0.381 | 2.382 | 88.402 | | | | | | |
| 9 | 0.298 | 1.865 | 90.267 | | | | | | |
| 10 | 0.295 | 1.845 | 92.113 | | | | | | |
| 11 | 0.284 | 1.773 | 93.886 | | | | | | |
| 12 | 0.235 | 1.468 | 95.353 | | | | | | |
| 13 | 0.220 | 1.377 | 96.730 | | | | | | |
| 14 | 0.194 | 1.210 | 97.940 | | | | | | |
| 15 | 0.190 | 1.189 | 99.129 | | | | | | |
| 16 | 0.139 | 0.871 | 100.000 | | | | | | |

Extraction Method: Principal Component Analysis. Rotation Method: Varimax with Kaiser Normalization. [a]. Rotation converged in 5 iterations. Source: Authors' computation in SPSS.

## Appendix C

**Table A3.** KMO and Bartlett's Test.

| Kaiser-Meyer-Olkin Measure of Sampling Adequacy. | | 0.916 |
|---|---|---|
| Bartlett's Test of Sphericity | Approx. Chi-Square | 5419.590 |
| | Df | 780 |
| | Sig. | 0.000 |

Source: Authors' computation in SPSS.

**Table A4.** Total Variance Explained.

| Component | Initial Eigenvalues | | | Extraction Sums of Squared Loadings | | | Rotation Sums of Squared Loadings | | |
|---|---|---|---|---|---|---|---|---|---|
| | Total | % of Variance | Cumulative % | Total | % of Variance | Cumulative % | Total | % of Variance | Cumulative % |
| 1 | 8.113 | 62.408 | 62.408 | 8.113 | 62.408 | 62.408 | 5.689 | 43.765 | 43.765 |
| 2 | 1.093 | 8.404 | 70.812 | 1.093 | 8.404 | 70.812 | 3.516 | 27.047 | 70.812 |
| 3 | 0.882 | 6.787 | 77.600 | | | | | | |
| 4 | 0.543 | 4.178 | 81.777 | | | | | | |
| 5 | 0.488 | 3.753 | 85.530 | | | | | | |
| 6 | 0.415 | 3.194 | 88.724 | | | | | | |
| 7 | 0.389 | 2.992 | 91.716 | | | | | | |
| 8 | 0.280 | 2.153 | 93.869 | | | | | | |
| 9 | 0.238 | 1.828 | 95.697 | | | | | | |
| 10 | 0.166 | 1.280 | 96.977 | | | | | | |
| 11 | 0.156 | 1.203 | 98.180 | | | | | | |
| 12 | 0.131 | 1.009 | 99.190 | | | | | | |
| 13 | 0.105 | 0.810 | 100.000 | | | | | | |

Extraction Method: Principal Component Analysis. Source: Authors' computation in SPSS.

## Appendix D

**Table A5.** KMO and Bartlett's Test.

| Kaiser-Meyer-Olkin Measure of Sampling Adequacy. | | 0.920 |
|---|---|---|
| Bartlett's Test of Sphericity | Approx. Chi-Square | 5095.677 |
| | Df | 105 |
| | Sig. | 0.000 |

Source: Authors' computation in SPSS.

**Table A6.** Total Variance Explained.

| Component | Initial Eigenvalues | | | Extraction Sums of Squared Loadings | | | Rotation Sums of Squared Loadings | | |
|---|---|---|---|---|---|---|---|---|---|
| | Total | % of Variance | Cumulative % | Total | % of Variance | Cumulative % | Total | % of Variance | Cumulative % |
| 1 | 7.828 | 52.186 | 52.186 | 7.828 | 52.186 | 52.186 | 6.414 | 42.757 | 42.757 |
| 2 | 1.824 | 12.160 | 64.346 | 1.824 | 12.160 | 64.346 | 3.238 | 21.588 | 64.346 |
| 3 | 0.922 | 6.147 | 70.492 | | | | | | |
| 4 | 0.727 | 4.847 | 75.339 | | | | | | |

<div align="center">**Table A6.** *Cont.*</div>

| Component | Initial Eigenvalues | | | Extraction Sums of Squared Loadings | | | Rotation Sums of Squared Loadings | | |
|---|---|---|---|---|---|---|---|---|---|
| | Total | % of Variance | Cumulative % | Total | % of Variance | Cumulative % | Total | % of Variance | Cumulative % |
| 5 | 0.671 | 4.473 | 79.812 | | | | | | |
| 6 | 0.577 | 3.844 | 83.656 | | | | | | |
| 7 | 0.443 | 2.956 | 86.612 | | | | | | |
| 8 | 0.411 | 2.743 | 89.355 | | | | | | |
| 9 | 0.358 | 2.389 | 91.744 | | | | | | |
| 10 | 0.268 | 1.790 | 93.534 | | | | | | |
| 11 | 0.242 | 1.616 | 95.150 | | | | | | |
| 12 | 0.220 | 1.466 | 96.616 | | | | | | |
| 13 | 0.201 | 1.341 | 97.958 | | | | | | |
| 14 | 0.179 | 1.193 | 99.151 | | | | | | |
| 15 | 0.127 | 0.849 | 100.000 | | | | | | |

<div align="center">Extraction Method: Principal Component Analysis. Source: Authors' computation in SPSS.</div>

## Appendix E

**Table A7.** KMO and Bartlett's Test.

| | | |
|---|---|---|
| Kaiser-Meyer-Olkin Measure of Sampling Adequacy. | | 0.865 |
| Bartlett's Test of Sphericity | Approx. Chi-Square | 3861.996 |
| | Df | 55 |
| | Sig. | 0.000 |

<div align="center">Source: Authors' computation in SPSS.</div>

**Table A8.** Total Variance Explained.

| Component | Initial Eigenvalues | | | Extraction Sums of Squared Loadings | | | Rotation Sums of Squared Loadings | | |
|---|---|---|---|---|---|---|---|---|---|
| | Total | % of Variance | Cumulative % | Total | % of Variance | Cumulative % | Total | % of Variance | Cumulative % |
| 1 | 5.651 | 51.375 | 51.375 | 5.651 | 51.375 | 51.375 | 4.256 | 38.692 | 38.692 |
| 2 | 2.004 | 18.215 | 69.589 | 2.004 | 18.215 | 69.589 | 3.399 | 30.898 | 69.589 |
| 3 | 0.853 | 7.758 | 77.347 | | | | | | |
| 4 | 0.593 | 5.394 | 82.741 | | | | | | |
| 5 | 0.544 | 4.945 | 87.686 | | | | | | |
| 6 | 0.356 | 3.236 | 90.922 | | | | | | |
| 7 | 0.321 | 2.917 | 93.839 | | | | | | |
| 8 | 0.257 | 2.334 | 96.173 | | | | | | |
| 9 | 0.179 | 1.627 | 97.800 | | | | | | |
| 10 | 0.135 | 1.228 | 99.028 | | | | | | |
| 11 | 0.107 | 0.972 | 100.000 | | | | | | |

<div align="center">Extraction Method: Principal Component Analysis. Source: Authors' computation in SPSS.</div>

## Appendix F

**Table A9.** Model Summary [b].

| Model | R | R Square | Adjusted R Square | Std. Error of the Estimate | Durbin-Watson |
|---|---|---|---|---|---|
| 1 | 0.506 [a] | 0.256 | 0.248 | 0.847 | 2.022 |

[a]. Predictors: (Constant), PC_1.1, PC_1.2, PC_1.3, PC_2.1, PC_2.2. [b]. Dependent variable: Q5.1. Source: Authors' computation in SPSS.

**Table A10.** ANOVA [b].

| | Model | Sum of Squares | Df | Mean Square | F | Sig. |
|---|---|---|---|---|---|---|
| | Regression | 113.837 | 5 | 22.767 | 31.763 | 0.000 [a] |
| 1 | Residual | 331.163 | 462 | 0.717 | | |
| | Total | 445.000 | 467 | | | |

[a]. Predictors: (Constant), PC_1.1, PC_1.2, PC_1.3, PC_2.1, and PC_2.2. [b]. Dependent variable: Q5.1. Source: Authors' computation in SPSS.

## Appendix G

**Table A11.** Model Summary [b].

| Model | R | R Square | Adjusted R Square | Std. Error of the Estimate | Durbin-Watson |
|---|---|---|---|---|---|
| 2 | 0.525 [a] | 0.276 | 0.268 | 0.901 | 2.104 |

Source: Authors' computation in SPSS. [a]. Predictors: (Constant), PC_1.1, PC_1.2, PC_1.3, PC_2.1, and PC_2.2. [b]. Dependent variable: Q5.2.

**Table A12.** ANOVA [b].

| | Model | Sum of Squares | df | Mean Square | F | Sig. |
|---|---|---|---|---|---|---|
| | Regression | 142.866 | 5 | 28.573 | 35.218 | 0.000 [a] |
| 2 | Residual | 374.826 | 462 | 0.811 | | |
| | Total | 517.692 | 467 | | | |

[a]. Predictors: (Constant), PC_1.1, PC_1.2, PC_1.3, PC_2.1, and PC_2.2. [b]. Dependent variable: Q5.2. Source: Authors' computation in SPSS.

## Appendix H

**Table A13.** Model Summary [b].

| Model | R | R Square | Adjusted R Square | Std. Error of the Estimate | Durbin-Watson |
|---|---|---|---|---|---|
| 3 | 0.430 [a] | 0.185 | 0.176 | 0.871 | 2.105 |

[a]. Predictors: (Constant), PC_1.1, PC_1.2, PC_1.3, PC_3.1, and PC_3.2. [b]. Dependent variable: Q5.3. Source: Authors' computation in SPSS.

**Table A14.** ANOVA [b].

| | Model | Sum of Squares | df | Mean Square | F | Sig. |
|---|---|---|---|---|---|---|
| | Regression | 79.801 | 5 | 15.960 | 21.016 | 0.000 [a] |
| 3 | Residual | 350.857 | 462 | 0.759 | | |
| | Total | 430.658 | 467 | | | |

[a]. Predictors: (Constant), PC_1.1, PC_1.2, PC_1.3, PC_3.1, and PC_3.2. [b]. Dependent variable: Q5.3. Source: Authors' computation in SPSS.

## Appendix I

**Table A15.** Model Summary [b].

| Model | R | R Square | Adjusted R Square | Std. Error of the Estimate | Durbin-Watson |
|---|---|---|---|---|---|
| 4 | 0.676 [a] | 0.457 | 0.451 | 0.797 | 2.147 |

[a]. Predictors: (Constant), PC_1.1, PC_1.2, PC_1.3, PC_3.1, and PC_3.2. [b]. Dependent variable: Q5.4. Source: Authors' computation in SPSS.

**Table A16.** ANOVA [b].

| | Model | Sum of Squares | df | Mean Square | F | Sig. |
|---|---|---|---|---|---|---|
| 4 | Regression | 247.129 | 5 | 49.426 | 77.753 | 0.000 [a] |
| | Residual | 293.683 | 462 | 0.636 | | |
| | Total | 540.812 | 467 | | | |

[a]. Predictors: (Constant), PC_1.1, PC_1.2, PC_1.3, PC_3.1, and PC_3.2. [b]. Dependent variable: Q5.4. Source: Authors' computation in SPSS.

## Appendix J

**Table A17.** Model Summary [b].

| Model | R | R Square | Adjusted R Square | Std. Error of the Estimate | Durbin-Watson |
|---|---|---|---|---|---|
| 5 | 0.561 [a] | 0.314 | 0.307 | 0.744 | 2.080 |

[a]. Predictors: (Constant), PC_1.1, PC_1.2, PC_1.3, PC_4.1, and PC_4.2. [b]. Dependent variable: Q5.5. Source: Authors' computation in SPSS.

**Table A18.** ANOVA [b].

| | Model | Sum of Squares | df | Mean Square | F | Sig. |
|---|---|---|---|---|---|---|
| 5 | Regression | 117.190 | 5 | 23.438 | 42.386 | 0.000 [a] |
| | Residual | 255.468 | 462 | 0.553 | | |
| | Total | 372.658 | 467 | | | |

[a]. Predictors: (Constant), PC_1.1, PC_1.2, PC_1.3, PC_4.1, and PC_4.2. [b]. Dependent variable: Q5.5. Source: Authors' computation in SPSS.

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
