# Peer review of "Perceptions and Attitudes of Generation Z Students towards the Responsible Management of Smart Cities"

_sustainability, doi:10.3390/su151813967_

Round 1
Reviewer 1 Report
Overall, paper is well written, research is adequately described. In my opinion it could be a bit shorter for readers to faster adopt the topic.
The conclusion could be improved by answering questions:
- what are the all-over weaknesses of the research;
- what could be future research expectations;
- what are the recommendations and implications of the research;
- how could/should futures studies improve the model etc.
Reviewer 2 Report
The paper aims to discuss the perceptions and the attitudes of the Generation Z towards the responsible management of smart cities.
First, I think that the abstract is too long. Please reduce its size to fit the journal's requirements.
Second, the introduction holds 7 pages, which is extremely long for a research paper. Please consider reducing the size of the introduction and keep in the paper only the relevant information related to the Generation Z and the importance of this generation to the now-a-days research focus.
Also, I do not find necessarily to dedicate a table to definitions - please add and discuss them as plain text.
Furthermore, in the methodology I think that the authors should discuss the questionnaire used, its questions (also to add the questionnaire as an appendix to the paper). I see no purpose for the provided formulas as the results of these formulas are determined through the use of a software in the paper and the authors do not compute them manually. Thus, please re-write the entire methodology section.
As for the results section (section 3), I think that the sample is not representative for the Generation Z: the authors mention in the introduction that the Z generation has between 11 and 26 years old, while in the paper the generation Z is represented only persons between 23 and 26 years old. As a result, there is a large part of generation Z which is not represented in the study, namely the 11- 23 years old persons. Furthermore, the selected respondents are all students of various economic studies - this ia also an under-representation of the persons between 21 and 23 years old - some of them might not follow a university education degree, while, from the other that have decided to go to the university, a large number of students might be of different specialization. Thus, given that the representativity is not met -- see the above reasons, I found that the rest of the analysis is in vain.
Minor English changes required.
Reviewer 3 Report
The name of the country is not appropriate to be listed as a keyword. I think "public perception" would be a better keyword for this paper.
The current introduction is too reduntant (6 pages). It is recommended to add some new sections like "literature review" and "theoretical background" et al.
The table 1 is really hard to be understanded. I recommend the authors to add more columns to make the table clearer.
Some newly published papers in the relevant fields of public perception of engineering projects (https://doi.org/10.3992/jgb.18.2.65) have not been introduced in the literature review section and listed in the reference list.
It is not needed to put all the data in the main text. Some of the tables are not needed to be put in the main text, and could be moved into the appendix, like table 15, 16 & 17 et al.
There is not so much information cotained in the table 24-26, and these three tables could be emerged into one. The authors should check the whole paper. The same kinds of problems also exist in other parts of the paper.
Sections names of the reults section should have definite meanings. It is not suggested to use data types as the section names, like "3.4. Regressive Analysis".
In a word, the paper is interesting but the current version is really too reduntant. The authors should delete the redundant parts re move these part to the appendix. So as to make the paper more reader-friendly.
The name of the country is not appropriate to be listed as a keyword. I think "public perception" would be a better keyword for this paper.
The current introduction is too reduntant (6 pages). It is recommended to add some new sections like "literature review" and "theoretical background" et al.
The table 1 is really hard to be understanded. I recommend the authors to add more columns to make the table clearer.
Some newly published papers in the relevant fields of public perception of engineering projects (https://doi.org/10.3992/jgb.18.2.65) have not been introduced in the literature review section and listed in the reference list.
It is not needed to put all the data in the main text. Some of the tables are not needed to be put in the main text, and could be moved into the appendix, like table 15, 16 & 17 et al.
There is not so much information cotained in the table 24-26, and these three tables could be emerged into one. The authors should check the whole paper. The same kinds of problems also exist in other parts of the paper.
Sections names of the reults section should have definite meanings. It is not suggested to use data types as the section names, like "3.4. Regressive Analysis".
In a word, the paper is interesting but the current version is really too reduntant. The authors should delete the redundant parts re move these part to the appendix. So as to make the paper more reader-friendly.
Reviewer 4 Report
In this paper, the authors begin with a brief introduction to smart cities, responsible management, and Generation Z.Then, they give the research gap: there are no studies regarding the relationships between these research areas. In order to clarify and analyse Generation Z’s perceptions and attitudes towards the responsible management of a smart city, the authors select all undergraduate students of the Faculty of Business Administration of the University of Bucharest as the subjects of the survey data. The results of this study show that the respondents are aware of the role played by the city government in ensuring responsible management towards the economic, social, and environmental issues of a smart city. Although I am not an expert in this field, I still understand the connotation of the study through the author's detailed description and careful derivation. Thus, I recommend this paper can be accepted for publication in Sustainability.
However, there are still some issues that need to be considered and improved before the official publication. Specific comments are as follows.
#1: There are many grammar and tense errors in the paper, which seriously affect the experience of the comments. I hope the author will give this a serious review.
#2: In table 1, the authors give some definitions of smart city. However, this does not give a clear connection in context and is particularly abrupt.
#3: Some abbreviations of the article do not give specific instructions, for instance IT&C. I think it will be very confusing for the readers.
#4: The article focuses on Gen Z's perceptions and attitudes towards the responsible management of smart cities, but, at least for me, I am puzzled why the author begin to spend a lot of ink on what smart cities are rather than what Gen Z is. I hope the author can clarify it.
#5: After carefully reviewing this manuscript, I wonder why the survey data only come from all undergraduate students of the Faculty of Business Administration of the University of Bucharest. Therefore, a question naturally arises as to whether such survey results are representative. Of course, I am not an expert in this area, and I hope we can have a discussion on this issue.
None
Round 2
Reviewer 2 Report
I thank the authors for considering the comments in the previous round of review and for the changes made in the paper in accordance to these comments. Much appreciated!
Reviewer 3 Report
The quality of the paper has been largely improved in this version, and I recommend to accept the paper in the current version.
Reviewer 4 Report
The authors have revised the manuscript according to the comments. The model design of this paper is novel and the results are interesting, which can attract the attention of relevant researchers. I recommend that it be accepted and published in its current version.